# *Withania somnifera* (L.) Dunal, a Potential Source of Phytochemicals for Treating Neurodegenerative Diseases: A Systematic Review

**DOI:** 10.3390/plants13060771

**Published:** 2024-03-08

**Authors:** Valentina Lerose, Maria Ponticelli, Nadia Benedetto, Vittorio Carlucci, Ludovica Lela, Nikolay T. Tzvetkov, Luigi Milella

**Affiliations:** 1Department of Science, University of Basilicata, V.le Ateneo Lucano 10, 85100 Potenza, Italy; valeroses90@gmail.com (V.L.); nadia.benedetto@unibas.it (N.B.); vittorio.carlucci@unibas.it (V.C.); ludovica.lela@unibas.it (L.L.); 2Institute of Molecular Biology “Roumen Tsanev”, Department of Biochemical Pharmacology & Drug Design, Bulgarian Academy of Sciences, Acad. G. Bonchev Str., Bl. 21, 1113 Sofia, Bulgaria; ntzvetkov@gmx.de

**Keywords:** *Withania somnifera* (L.) Dunal, withanolides, Alzheimer’s disease, Parkinson’ s disease, ALS disease, Huntington’s disease

## Abstract

*Withania somnifera* (L.) Dunal is a medicinal plant belonging to the traditional Indian medical system, showing various therapeutic effects such as anti-cancer, anti-inflammatory, anti-microbial, anti-diabetic, and hepatoprotective activity. Of great interest is *W. somnifera*’s potential beneficial effect against neurodegenerative diseases, since the authorized medicinal treatments can only delay disease progression and provide symptomatic relief and are not without side effects. A systematic search of PubMed and Scopus databases was performed to identify preclinical and clinical studies focusing on the applications of *W. somnifera* in preventing neurodegenerative diseases. Only English articles and those containing the keywords (*Withania somnifera* AND “neurodegenerative diseases”, “neuroprotective effects”, “Huntington”, “Parkinson”, “Alzheimer”, “Amyotrophic Lateral Sclerosis”, “neurological disorders”) in the title or abstract were considered. Reviews, editorials, letters, meta-analyses, conference papers, short surveys, and book chapters were not considered. Selected articles were grouped by pathologies and summarized, considering the mechanism of action. The quality assessment and the risk of bias were performed using the Cochrane Handbook for Systematic Reviews of Interventions checklist. This review uses a systematic approach to summarize the results from 60 investigations to highlight the potential role of *W. somnifera* and its specialized metabolites in treating or preventing neurodegenerative diseases.

## 1. Introduction

Ayurveda, a popular Indian system of medicine, has a well-developed line of action for managing and treating brain-associated disorders. A list of about 450 Ayurvedic medicinal plants, 56 popular plants, or their active metabolites is available for neurological disorders in Ayurvedic prescriptions [1]. One of the traditional, well-known Indian medicinal plants is *Withania somnifera* (L.) Dunal, which is marketed as a common ingredient in several Ayurvedic formulations for the treatment of neurological disorders. *Withania somnifera*, popularly known as Ashwagandha in Sanskrit, has been used extensively in herbal medicine since its introduction in 6000 BC; it is an evergreen woody shrub of the Solanaceae family and has been given several names with different meanings, such as “Indian winter cherry” or “Indian ginseng” in English, “Punir” or “Asgandh” in Hindi, and “Asgand’ in Urdu [2]. The species name “*somnifera*” means “sleep inducer” in Latin, due to its amazing anti-stress properties, while the popular name “Ashwagandha” comes from “ashwa” (horse) and “gandha” (smell), as the roots have a characteristic “wet horse” smell [3]. In Ayurvedic literature, it is classified as a rejuvenating agent, since it promotes both physical and mental health, revitalizes the body in an incapacitated state, and maintains longevity. *W. somnifera* is potentially useful for many neurological disorders such as epilepsy, Alzheimer’s disease, Parkinson’s disease, cerebral ischaemia, and tardive dyskinesia [4]. In addition to its neurological effects, *W. somnifera* possesses several other pharmacological properties, such as anti-inflammatory, anti-diabetic, cardioprotective, anti-hepatitis, anti-osteoporotic, and anti-neoplastic activities [5,6]. In traditional medicine, *W. somnifera* has been used as an anti-microbial and aphrodisiac, as well as to treat weight gain, thyroid dysfunction, and insomnia [7]. The plant is an erect, greyish, evergreen shrub, 0.5–2 m tall, with long tuberous roots, short stems, oblong, stalked leaves, and greenish, bisexual flowers borne in the axils; the fruits are globose berries, 6 mm in diameter, orange-red when ripe, and enclosed in an inflated, membranous, persistent calyx. It is mainly found in regions from the Canary Islands and the Mediterranean to the tropical regions of South Asian countries (India and Sri Lanka), some Middle Eastern countries, China, and Africa, and it is grown in gardens in the warmer parts of Europe. In the Indian subcontinent, however, it is cultivated as a medicinal plant mainly for its fleshy roots, which are widely used in Unani and Ayurvedic systems of medicine, as they contain a wealth of bioactive compounds with several pharmacological implications [8,9,10]. In addition to the roots, the leaves are useful in treating fever and swelling, the flowers have been used as an astringent and diuretic, the fruits are used to treat skin ulcers, tumours, and carbuncles, and the seeds are effective in increasing sperm count and testicular growth [11]. *W. somnifera* has been shown to prevent neurotoxic and neurodegenerative conditions by exerting anti-oxidant mechanisms [3].

Several investigations have demonstrated that neuronal cells are vulnerable to oxidative damage due to their high polyunsaturated fatty acid content in membranes and the brain’s elevated oxygen consumption. Higher concentrations of reactive oxygen species and lipid peroxidation are associated with oxidative stress, disrupting the balance between pro-oxidant and anti-oxidant levels and leading to eventual neuronal loss [12,13,14]. Although the exact molecular pathogenesis of neurodegeneration due to the dysregulation of pro- and anti-oxidant mechanisms remains elusive, numerous potent anti-oxidants have demonstrated great potential in modifying disease phenotypes [15]. *W. somnifera* extract and its active constituents, as shown in the rest of this article, have been demonstrated to possess anti-oxidant properties and rescue neuronal cells from toxins by regulating different anti-oxidant enzymes, amyloid beta clearance, calcium influx, neurite outgrowth, lipid peroxidation, and inflammation, and similar debilitating processes that are involved in Alzheimer’s, Parkinson’s, Amyotrophic lateral sclerosis (ALS), and Huntington’s disease (Figure 1).

The rationale of this review is to explore the research articles (comprising *in vitro*, *in vivo*, and clinical investigations) available to date to provide a comprehensive overview of *W. somnifera* constituents’ activity in preventing and treating neurodegenerative diseases like Alzheimer’s, Parkinson’s, Amyotrophic lateral sclerosis (ALS), and Huntington’s disease, using a systematic approach. Specifically, this review aims to screen the dosages used and the results obtained in preclinical and clinical practice to inspire novel research approaches and enhance *W. somnifera* curative application.

### 1.1. Phytochemistry of W. somnifera

The phytochemical profile of *W. somnifera* has been extensively studied, thus identifying various specialized metabolites that may differ based on the tissue type, growth conditions, and variety of *W. somnifera* [16]. It has been reported that 62 major and minor primary and secondary metabolites are present in leaves and 48 in roots, of which 29 are found in both leaves and roots [17]. The bioactive constituents of *W. somnifera* reported in the literature include steroidal flavanol glycosides, glycowithanolides, steroidal lactones, and phenolics [17,18]. Until now, more than 12 alkaloids, around 40 withanolides, and several sitoindosides have been reported in the aerial parts, roots, and berries of *W. somnifera* [19]. The leaves are reported to contain five unidentified alkaloids (yield, 0.09%), 12 withanolides, many free amino acids, chlorogenic acid, glycosides, glucose, condensed tannins, and flavonoids [20]. Moreover, the roots contain alkaloids, amino acids, steroids, volatile oil, starch, reducing sugars, glycosides, hentriacontane, dulcitol, and withaniol [21]. Table 1 lists the principal active molecules of *W. somnifera*.

#### 1.1.1. Withanolides

Withanolides are a unique class of steroidal lactones. Over 130 withanolides are present in 15 genera of the Solanaceae family. According to some studies, withanolides have been isolated from some other Leguminosae and Labiatae families, as well as from some marine organisms [9]. *W. somnifera* plants produce the largest number of withanolides with a highly diversified functional group. Withanolides are *W. somnifera*’s key biochemicals, ranging from 0.001% to 0.5% dry weight and predominantly concentrated in leaves and roots [25]. Withanolides are formed by 28-carbon-containing natural steroidal lactone built on an ergostane skeleton, in which C-22 and C-26 are oxidized to form six- or five-membered lactone rings. The elementary structure, the 22-hydroxy ergostane-26-oic acid 26, 22-lactones, is labelled the withanolide skeleton [26]. Many structures have also been isolated from *W. somnifera*, which are the modifications or structural variants of withanolides with modifications of either the carbocyclic skeleton or side chain. These compounds are polyoxygenated; hence, it is thought that plants able to elaborate them possess an enzyme system capable of oxidizing all the carbon atoms in a steroid nucleus [9]. Although structurally similar, withanolide A differs from withanolide B through the presence of an additional hydroxyl group on the C-20 atom, and both can inhibit amyloid beta aggregation, known to cause Alzheimer’s disease [27,28]. 4*β*,27-dihydroxy-1-oxo-5*β*, 6*β*-epoxywitha-2-24-dienolide, also known as withaferin A, was first isolated from the *W. somnifera* leaves of a South Asian variety at a concentration of 0.13–0.31% dry weight. The quantitative analysis of Indian chemotypes of *W. somnifera* by TLC densitometry and HPLC analysis showed that withaferin A was present at a concentration of 1.6% in leaves, while it is present at very low concentrations in roots and stems [29,30]. Withanone is another withanolide found in significant amounts in Ashwagandha extracts (19 and 3 mg/g dry weight in leaves and roots, respectively) and is thought to be responsible for many of the medicinal properties of *W. somnifera* such as anti-oxidant, anti-amyloid, and anti-inflammatory activity [31].

#### 1.1.2. Withanolide Glycosides

##### Withanamides

Withanamides A-I are purified from the methanolic extract of the fruits of *W. somnifera*. Their structure contains a 5-hydroxytryptamine (serotonin) base with hydroxyfatty amide and diglucosidic moieties. The structural difference between withanamide A and withanamide C is only in the fatty acid side chain, as withanamide A has two double bonds in its side chain compared to withanamide C (Table 1). Withanamide A and withanamide C help to neutralize the toxicity of *β*-amyloid protein (BAP) and protect the cells from cell death. Withanamide A has been shown to represent a potential molecule usable for treating Alzheimer’s disease [32]. Withanamides have antioxidant properties, as their ability to inhibit lipid peroxidation (LPO) is equal to or better than the commercial antioxidants BHA, BHT, and TBHQ and far better than serotonin. Their potent antioxidant activity is probably due to the hydroxylated long-chain acyl group [24].

##### Withanosides

*W. somnifera* also contains withanolide glycosides or glycowithanolides known as withanosides with mostly a 6-O-*β*-d-glucopyranosyl-β-d-glucopyranosyl type of glycosidic linkage (Table 1). Seven withanolide glycosides named withanosides I, II, III, IV, V, VI, and VII have been isolated from the roots of *W. somnifera* [23]. Withanosides differ from withanolide in the presence of an additional hydroxyl group at the C-20 and C-27 atoms. Although structurally similar, withanoside IV differs from withanoside V in the presence of an extra hydroxyl group on the C-27 atom. Withanoside IV and withanoside V were seen to reduce the formation of the amyloid β-peptide fibril responsible for Alzheimer’s disease [27,28], while withanosides IV and VI induced dendritic formation [33]. A bioassay-guided purification of the methanolic extract of *W. somnifera* fruits yielded withanosides IV, V, and VI. These compounds were then tested for their ability to inhibit lipid peroxidation in a model system using large unilamellar vesicles, and only withanoside V inhibited lipid peroxidation by 82% at a concentration of 10 μg/mL, whereas the inhibition of withanoside IV was only 25% at 100 μg/mL and that of withanoside VI was 86% at 50 μg/mL; hydroxylation at C-27 in withanoside IV decreased the activity more than the hydroxylation at C-20 in withanoside VI. This may be due to hydrogen bonding between the C-27 hydroxyl and the carbonyl group of the lactone moiety in withanoside IV [24].

#### 1.1.3. Sominone

Withanoside IV is a steroidal saponin conjugated with two glucose residues at the C3 position and can be metabolized by enterobacterial *β*-glucosidases to a sapogenin, sominone, which has been identified as the major metabolite in the serum after the oral administration of withanoside IV. Sominone was first isolated from the methanol extract of the whole plant of *W. somnifera*, although the content was not reported [34]. However, even though Kuboyama, Tohda, and Komatsu [22] checked for sominone in the methanol extract of Ashwagandha, no presence was detected. Sominone is indeed a metabolite of withanoside IV and is not present in Ashwagandha as a natural constituent. It was enzymatically prepared from withanoside IV treated with naringinase (an enzyme with the dual activity of *α*-l-rhamnosidase and *β*-d-glucosidase) for 4 days at 37 °C in 8 mL phosphate–citrate buffer (pH 5.25) [22] (Figure 2).

Sominone was shown to induce axonal and dendritic regeneration and synaptic reconstruction in cultures of rat cortical neurons damaged by the amyloid peptide Aβ (25–35) [22].

## 2. Results and Discussion

### 2.1. Study Characteristics

This systematic review followed the PRISMA guidelines [35] and included articles published between 2000 and 2023. The search was conducted using Scopus [http://www.scopus.com (accessed 15 December 2023)] and PubMed [http://www.ncbi.nlm.nih.gov/pubmed (accessed 15 December 2023)] as databases and comprises all reports published until January 2024. For the search, the keywords used include the species *Withania somnifera* (L.) Dunal paired with the following words: “neurodegenerative diseases”, “neuroprotective effects”, “Huntington”, “Parkinson”, “Alzheimer”, “Amyotrophic Lateral Sclerosis”, and “neurological disorders” (view Materials and Methods, Section 3.1, for more details). Only English-language publications were considered. The initial selection provided 960 articles, of which 407 were found on PubMed and 553 on Scopus. Among the 960 items, only 300 were duplicates, and 660 were screened. Of those 660 articles, 116 fitted with the inclusion criteria, while 544 were off-topic based on the exclusion criteria. After a full text examination of the 116 selected articles, 58 were excluded due to the absence of topics of interest. The final reference list comprises 60 items, of which 2 publications come from other sources (Figure 3).

The selected papers originate from eight countries (Figure 4a), among which India is the most representative. This systematic review included 57 *in vitro/vivo* reports and 3 clinical investigations (Figure 4b). For *in vivo* and clinical investigations, dosages, administration frequency, and therapy duration were cited in every case. Results from the risk bias assessment, founded on a checklist from the Cochrane Handbook for Systematic Reviews of Interventions, are reported in Figure 4c. The lack of clinical studies with a low risk of bias is related to the evidence that the selected trials involve a limited number of individuals and short treatment times. In addition, selected clinical trials predominantly assess cognitive function in healthy subjects, indicating a potential risk of bias in future application in patients with neurodegenerative diseases.

### 2.2. Whitania somnifera (L.) Dunal and Alzheimer’s Disease

Alzheimer’s disease (AD) is the most common neurological disease (ND) and the leading cause of dementia. Classic AD symptoms include dementia, memory and spatial problems, movement disorders, depression, and hallucinations. Patients also experience anomic aphasia, acalculia, and apathy [37,38]. Many phenomena that lead to AD remain unclear, although significant advances have been made in research to explain AD-related changes. The most widely accepted hypothesis is the so-called amyloid cascade. It comprises amyloid beta (A*β*) accumulation due to genetic defects (i.e., APP, presenilin 1, presenilin 2), environmental factors, and other stressors. These senile plaques induce an immune response, leading to inflammation and tau hyperphosphorylation and aggregation into tangles [39] with consequent neuronal degeneration, death, and impaired neurotransmission in many brain regions [40]. There are two main symptomatic treatments for AD: acetylcholinesterase inhibitors (e.g., donepezil, galantamine, and rivastigmine), which increase the availability of acetylcholine at synapses; and the non-competitive N-methyl-d-aspartate (NMDA) receptor antagonist (memantine), which modulates the opening of calcium channels and aims to reduce the excitatory neurotoxicity of l-glutamate without interfering with its physiological actions [41].

#### 2.2.1. Effect of *W. somnifera* on Axon Degeneration and Synaptic Reconstruction

In Alzheimer’s disease, A*β* deposit formations in the brain are widely recognized as a critical cause of axonal atrophy and synaptic degeneration [42,43,44]. Promoting axonal and synaptic regeneration may lead to the reconstruction of neuronal networks and fundamental recovery from Alzheimer’s disease. Therefore, the effects of the methanol extracts of Ashwagandha on neurite outgrowth were investigated using an *in vitro* culture system. The methanol extract of Ashwagandha showed neurite outgrowth-promoting activity in human neuroblastoma SK-N-SH cells at a concentration of 5 µg/mL for 5 days. Ashwagandha extract showed elongated multipolar neurites. The mRNA levels of the dendritic markers microtubule-associated protein 2 (MAP2) and postsynaptic density protein-95 (PSD-95) were measured by RT-PCR and found to be significantly increased by treatment with Ashwagandha extract. However, those of the axonal marker tau were not [45]. Withanolide A (WL-A), withanoside IV, and withanoside VI were identified as active constituents of the methanol extract that induced neurite outgrowth in human neuroblastoma SH-SY5Y cells and rat cortical neurons [33,46]. Double immunostaining was performed in rat cortical neurons using antibodies against phosphorylated NF-H (phosphorylated neurofilament-H) as an axonal marker and against MAP2 as a dendritic marker. In withanolide A-treated cells, the length of NF-H-positive processes was significantly increased compared to vehicle-treated cells (0.1% dimethylsulfoxide DMSO), whereas withanosides IV and VI increased the length of MAP2-positive processes. These results suggest that axons are predominantly elongated by withanolide A, while dendrites are predominantly elongated by withanosides IV and VI at a concentration of 1 µM each [33,46]. An *in vitro* axonal atrophy model was established using an active partial fragment of A*β*, A*β* (25–35), which induced axonal atrophy in a rat cortical neuron treated with 10 µM A*β* (25–35) for 4 days as potently as full-length A*β* (1–42) [47]. It was seen that withanolide A, withanoside IV, and withanoside VI induced axonal growth even in the presence of A*β* (25–35) at a concentration of 1 µM [22,47]. It is crucial to determine whether regenerated neurites are also able to reconstruct synapses. Withanolide A, withanoside IV, and withanoside VI have been shown to regenerate axons and dendrites, and, in particular, withanolide A promotes presynaptic formation *in vitro* compared to withanoside IV and withanoside VI [33]. Withanolide A has no glycoside, whereas withanoside IV and withanoside VI are glycosides. It has been speculated that withanolide compounds that do not have a glucose residue promote more presynaptic formation [33]. Using *in vitro* assay, Kuboyama et al. (2005) [47] and Kuboyama et al. (2006) [22] tested the effects of withanolide A, withanoside IV, and its metabolite sominone on pre- and postsynaptic maturation. *In vitro* rat cortical neurons were treated with 10 µM A*β* (25–35) for 4 days, resulting in a loss of pre- and postsynaptic densities [22,47]. Four days after adding A*β* (25–35), the cells were treated with sominone (1 µM), withanoside IV (1 µM), withanolide A (1 µM), NGF (100 ng/mL), or vehicle. Treatment with withanolide A significantly increased synaptophysin and PSD-95 expressions; also, withanoside IV and sominone significantly increased synaptophysin and MAP2 expressions compared to treatment with the vehicle. The number of presynapses was increased more by sominone treatment than withanoside IV at the same dose [47], suggesting that structural changes during deglycosylation may affect the potency of synaptogenesis.

*In vivo*, A*β* (25–35) (25 nmol) was injected intracerebroventricularly (i.c.v.) into the brains of male ddY mice [22,47]; the density of axons and synapses in the parietal cortex was reduced and the spatial memory of the mice was impaired. The oral administration of withanolide A and withanoside IV increased the density of axons and synapses in mice parietal cortices and improved the spatial memory deficit (water maze test) [22,47]. Following the oral administration of withanoside IV to mice, serum was collected and analysed by liquid chromatography–mass spectrometry [22]. As a result, withanoside IV was not detected in the serum, whereas sominone, an aglycon of withanoside IV, was detected [22]. Withanoside IV conjugates two glucoses at position C3; therefore, it was suggested that orally administered withanoside IV was deglycosylated at position C3 by intestinal bacteria. Sominone induced axonal regeneration and synaptic reconstruction in A*β* (25–35)-treated cortical neurons *in vitro* [22,48], and when 10 μmol/kg sominone was administered directly intraperitoneally for 9 days to the 5XFAD mouse model of AD, it improved object recognition memory compared with vehicle control treatment. Sominone treatment significantly increased axonal density in the frontal and parietal cortex but did not affect amyloid plaque deposition and activated microglia [48]. These results suggest that after oral administration, withanoside IV is metabolized to its active ingredient, sominone, which subsequently induces a marked recovery of neurites, synapses, and memory. These data suggest that withanoside IV and its metabolite sominone are potential drugs for Alzheimer’s disease. Tohda and Joyashiki (2009) [49] clarify why and how sominone was able to extend axons and dendrites. The target of sominone was identified as RET (rearranged during transfection), a receptor of the glial cell line-derived neurotrophic factor (GDNF) family with transmembrane tyrosine kinase activity, which was shown to be phosphorylated after sominone [49]. Phosphorylation was increased in cultured cortical neurons at least 15 min after sominone administration. The experimental knockdown of RET inhibited sominone-induced axonal growth in cultured cortical neurons, and GDNF secretion was not affected by sominone treatment. Sominone induced RET phosphorylation and outgrowth of axons and dendrites to a similar extent as GDNF (100 ng/mL); it may enhance the morphological plasticity of neurons by activating the RET pathway independently of GDNF and may be an exogenous low-molecular-weight stimulator of the RET pathway and/or a novel modulator of RET signalling [49]. Other research groups have reported withanolide A effectiveness on experimental models of Alzheimer’s disease. Patil et al. (2010) [50] reported that withanolide A at a concentration of 100 µM decreased beta-site amyloid precursor protein cleaving enzyme 1 (BACE1; known as *β*-secretase) expression and increased disintegrin and metalloproteinase domain-containing protein 10 (ADAM10; known as α-secretase) expression in cultured normal rat cortical neurons. A*β* is produced from amyloid precursor protein (APP) by processing with BACE1 and presenilin 1 (PS1; known as *γ*-secretase) [51]. When ADAM10 processes APP, A*β* production is attenuated, while non-toxic soluble APP*α* is alternatively produced. As a result, withanolide A increased soluble APP*α* production in cultured neurons (Figure 5).

Withanolide A was also shown to increase the expression of insulin-degrading enzyme (IDE), a key proteolytic enzyme involved in A*β* degradation, in primary rat cortical neurons. The activity, as well as the mRNA and protein levels of IDE, are decreased in the AD brain, and this decrease is associated with increased levels of A*β* compared to healthy controls [50]. These results suggest that withanolide A may reduce A*β* by increasing soluble APP*α* production and A*β* clearance. In view of these reports, withanolide A is an important candidate as a multifunctional drug against Alzheimer’s disease.

Recently, the methanolic extract of *W. somnifera* roots was shown to improve cognition via regulation of Na^+^–Ca^2+^ exchanger isoform 3 (NCX3) by restoring baseline Ca^2+^ levels after depolarisation in neurons of 4-month-old 5xFAD mice [52]. The aggregation of A*β* (1–42) plaques in the brain is likely caused by the aberrant proteolysis of amyloid precursor protein (APP), and the plaques subsequently trigger calcium influx-related neuronal damage, including mitochondrial dysfunction in AD [53]. Before plaques generation, small forms of A*β* (1–42), called oligomers, appear to be toxic to neurons by binding to neuronal plasma membrane proteins, such as NCX3, and disrupting intraneuronal Ca^2+^ homeostasis in AD. Thus, the toxic A*β* (1–42) oligomers accumulate to form plaques, which are thought to be responsible for Alzheimer’s disease symptoms. The immunoblotting analysis of the expression level of NCX3 indicated that treatment with *W. somnifera* significantly increased the expression of NCX3 in the cortex and hippocampus [52]. This suggests that the oxidative-stress-normalising effects of *W. somnifera* constituents, which act on the NCX3, may be therapeutic in improving cognition in Alzheimer’s disease.

#### 2.2.2. Neuroprotective Effects of *W. somnifera*

Kurapati et al. (2013–2014) [54,55] investigated the effects of methanol–chloroform (3:1) extract of Ashwagandha and its major constituent withanolide A against A*β* (1–42)-induced toxicity in cultured human neuroblastoma SK-N-MC cells. The *β*-amyloid treated cells (5 µM) showed cytotoxic effects with decreased cell growth compared to the control. However, when *W. somnifera* extract or withanolide A was added to the *β*-amyloid treated cultures at concentrations of 0.15 µg/mL and 1.25 µg/mL–20 µg/mL, respectively, the cytotoxic effects of *β*-amyloid were neutralized and the cells were comparable to plain and Ashwagandha treated controls, suggesting the chemopreventive or protective effects of Ashwagandha and withanolide A against *β*-amyloid induced toxicity [54,55]. Recently, the same effects have been reported by Dubey et al. (2021) [28] with withanolide A, withanolide B, withanoside IV, and withanoside V, which have significant therapeutic potential against amyloid *β* aggregation-induced cell toxicity, as the number of live cells was increased when these compounds were added (with half the concentration of the IC_50_ value) after the addition of A*β*. The cell culture assays also indicate that the ability of the derivatives to reduce the cytotoxic effects of amyloid *β* fibrillation could be mediated by free radicals and oxidative stress [28]. Cells treated with amyloid-*β* showed an increase in ROS production. A significant reduction in ROS production can be seen when the cells were treated with withanolide A, withanolide B, withanoside IV, and withanoside V. However, withanolide A and withanolide B reduced ROS production more than withanoside IV and withanoside V [28], suggesting they have the highest antioxidant activity. The difference between the withanolides and withanosides is the C-28 and C-40 steroidal lactones, with a six-membered lactone ring formed by the oxidation of the C-22 and C-28 on the ergostane backbone, which may be the reason why withanolide A and withanolide B are more effective in reducing ROS. Various transgenic models and *in vitro* studies suggest that intraneuronal accumulation of *β*-amyloid is an early event and plays an important role in the pathogenesis of AD [56,57]. To understand the effect of Ashwagandha on *β*-amyloid1-42 internalisation in SK-N-MC cells, cells were pre-incubated with methanol–chloroform (3:1) extract of Ashwagandha (0.15 µg/mL) for 3 h and then exposed to *β*-amyloid (1–42) for 16 h [54]. Cells were then stained with Congo red, a metachromatic anionic dye specific for *β*-amyloid; cultures treated with β-amyloid (1–42) alone showed much greater internalisation of the toxic peptide than when cells were incubated with *β*-amyloid (1–42) plus Ashwagandha [54,55]. The peroxisome proliferator-activated receptor (PPAR) is a ligand-activated nuclear receptor, and its activation is associated with the clearance of *β*-amyloid and the amelioration of pathological and behavioural deficits in Alzheimer’s disease [58]. Therefore, to understand the role of peroxisome proliferator-activated receptors (PPARs) during *β*-amyloid exposure as well as in the combination of *β*-amyloid plus Ashwagandha, Western blotting analysis was performed on cell lysates. The treatment of SK-N-MC cells with *β*-amyloid resulted in decreased PPAR*γ* protein levels (*p* < 0.01) compared to untreated control cells. In Ashwagandha plus *β*-amyloid-treated cells, no significant reduction in protein levels was observed compared to the untreated control. However, a relatively small decrease in PPAR*γ* levels was also observed in cultures treated with Ashwagandha alone [54]. Thus, the upregulation of PPAR*γ* by the extract supports neuroprotective effects against A*β*, but a causal relationship between PPAR*γ* expression and neuroprotective effects has not been established. Some studies suggest that free radicals and oxidative stress may mediate one of the pathways of *β*-amyloid-induced cytotoxicity. In this context, Kumar et al. (2010) [59] investigated the protective effects of an aqueous root extract of *W. somnifera* against H_2_O_2_ and *β*-amyloid (1–42)-induced cytotoxicity in differentiated PC12 (dPC12) cells [59]. A*β* has been shown to increase intracellular hydrogen peroxide (H_2_O_2_) levels. The aqueous root extract of *W. somnifera* significantly protected PC12 cells from H_2_O_2_-induced toxicity when cells were pre-incubated with *W. somnifera* extract for 24 h before exposure to H_2_O_2_ (200 µM), with extract concentrations of 50 and 100 µg/mL showing the greatest improvement over the negative control. At the highest extract concentration (200 μg/mL), the cytoprotective effects were abolished [59], possibly due to the direct cytotoxic and anti-proliferative properties of *W. somnifera* that are expressed at high concentrations [60]. Transmission electron microscopy was used to assess the inhibitory effect of the aqueous root extract of *W. somnifera* on A*β* fibril formation. A*β* peptide (50 μg/mL) was incubated for 24 h at 37 °C in the presence of water alone or with cholesterol (0.2 mg/mL), which is known to promote A*β* fibrillogenesis [61,62]. When A*β* peptide (50 μg/mL) was co-incubated with increasing concentrations of *W. somnifera* aqueous extract (6.25–50 μg/mL), fibril formation was significantly inhibited [63]. In addition, the morphology of the fibrils formed with the test samples appeared shorter in length than the control, indicating the anti-amyloidogenic activity of *W. somnifera* extract [63]. The inhibition of fibrillogenesis by *W. somnifera* root extract, demonstrated in this study, can be correlated with the neuroprotective activity of *W. somnifera* extract demonstrated against A*β*-induced cytotoxicity in cultured PC12 cells [32,59]. The only preliminary interpretation that can be made is that the chemical compounds present in the aqueous extract of *W. somnifera* roots may bind to hydrophobic regions of the A*β* peptide and, thus, inhibit A*β* fibril formation. Dubey et al. (2021) [28] used bioinformatics tools to study the binding and molecular stability of derivatives of *W. somnifera* with amyloid *β* and observed that these molecules interacted at the LVFFA [A*β* (17–21)] region, the inhibition site of amyloid-*β* (1–42). These studies revealed that the withanolides and withanosides interact with the hydrophobic core of amyloid-*β* (1–42) at the oligomeric stage, preventing further interaction with the monomers and reducing aggregation [28]. Jayaprakasam et al. (2004) [24] found a novel class of compounds from the fruit of *W. somnifera* with potent lipid peroxidation (LPO) inhibitory activity: the withanamides, which showed greater antioxidant capacity than the commercial antioxidants butylated hydroxytoluene (BHT), butylated hydroxylanisole (BHA), and ter-butylhydroxyhydroquinone (TBHQ) [24]. *In vitro*, Jayaprakasam et al. (2010) [32] tested two major withanamides A (WA) and C (WC), obtained from the methanolic fruit extract of *W. somnifera*, for their ability to protect PC-12 cells, rat neuronal cells, from *β*-amyloid (25–35)-induced cell damage. WA and WC were tested at a concentration of 50 μg/mL to determine their effect on protecting PC-12 cells from *β*-amyloid protein (A*β*) damage. A*β* at 10 μg/mL caused about 20% growth inhibition of PC-12 cells. Of the two compounds tested, WA was the most active and completely neutralised the toxic effect of A*β* on PC-12 cells. WC also protected the cells from A*β*-induced cell death and increased survival by 8% compared to the A*β* control [32]. Since WA was the most active of the withanamides tested, only the dose-dependent effect of WA on A*β*-induced toxicity was evaluated. The concentrations tested were 100, 50, 25, 12.5, and 6.25 μg/mL. Cells treated at 100 and 50 μg/mL showed 100% cell survival, indicating that at these concentrations, WA completely protected PC-12 cells from BAP-induced cell death [32]. The structure of the withanamides contained a serotonin (5-hydroxytryptamine) base with hydroxy fatty amide and diglucosidic moieties. For WA and WC, the structural variation is only in the fatty acid side chain. The molecule of WA had two double bonds in its side chain compared to WC [24]. The double bonds in WA provided a model that could be precisely superimposed on the active motif of A*β* (25–35) and was consistent with its higher protective effect observed in *in vitro* cell culture. The saturated fatty amide moiety in WC also paired well with most of the reactive motif in A*β*, protecting cells from A*β* insult [32], but the fact that it does not span the entire active motif explains why WA is stronger than WC in *in vitro* binding studies. Since it is known that WA and WC bind to A*β* and inhibit fibril formation, it is most likely that the conditions under which this occurs favoured the hydrophobic interaction between WA/WC and A*β* rather than hydrogen bonds leading to fibril formation. The hydrophobic interaction between A*β* and withanamides may also be enhanced by the presence of the serotonin moiety in WA and WC. The binding or association of WA or WC to the active motif of A*β* suggests that withanamides can prevent A*β* fibril formation [32]. Recently, the neuroprotective role of withaferin A against A*β* secretion and aggregation *in vitro* was also investigated for the first time [64]. SH-APP cells, a human neuroblastoma cell line stably overexpressing human APP751, were treated with different concentrations of withaferin A (0.5–10 µM); results showed that 2 µM withaferin A significantly reduced secreted A*β* (1–40) in SH-APP cells compared to the untreated control. Furthermore, the results were confirmed by flow cytometry and showed a dose-dependent reduction in A*β* (1–40) levels; the maximum reduction was reported at 2 µM withaferin A concentration without causing cellular toxicity [64]. Atluri et al. (2020) [65] investigated the effect of withaferin A on A*β* production by staining cells with Congo red (CR). SH-APP cell cultures treated with withaferin A (1 µM) alone showed less staining with the toxic A*β* peptide than SH-APP cell controls treated with dimethyl sulfoxide (DMSO). CR labelled the A*β* peptide and stained it red in DMSO-treated control SH-APP cells expressing higher levels of A*β*. Withaferin A-treated cells had lower levels of A*β*, and, therefore, less staining was observed in these treatment group cells [65]. This study is consistent with previous results that withaferin A has the ability to reduce A*β in vitro* [64].

#### 2.2.3. Anti-Neuroinflammation Activity of *W. somnifera*

Neuroinflammation is an important pathological feature in the progression of Alzheimer’s disease and associated neurocognitive impairment. Neuroinflammation is stimulated by damaged neurons and the deposition of insoluble A*β* peptides and neurofibrillary tangles (NFTs) [66]. Microglia, the phagocytes of the central nervous system, are key players in the maintenance and plasticity of neuronal circuits and are also involved in synapse protection and remodelling. In Alzheimer’s disease, microglia are stimulated by binding to soluble A*β* oligomers and A*β* fibrils through cell surface receptors, which induce an inflammatory response by activating the nucleotide-binding oligomerisation domain (NOD), leucine-rich repeat (LRR) and pyrin domain containing 3 (NLRP3) inflammasomes, and the nuclear factor kappa light chain enhancer of activated B cells (NF-kB) pathway, resulting in the release of proinflammatory cytokines and chemokines [67,68,69,70,71]. Microglia engulf A*β* fibrils by phagocytosis and degrade them using proteases such as neprilysin and insulin-degrading enzymes. In AD patients, the activation of the NLRP3 and NF-kB cascades inhibits A*β* phagocytosis by microglia, leading to the increased deposition of A*β* fibrils, creating a self-perpetuating loop that further induces neuroinflammation [71]. Based on reports on the role of withaferin A in inhibiting NF-kB-mediated neuroinflammation [72,73], Alturi et al. (2020) [65] used SH-APP and microglia mixed cell culture to investigate the anti-neuroinflammatory activity of WA. SH-APP cells and microglial cell lines (CHME5) were co-cultured for 48 h and incubated with different concentrations of withaferin A. After 48 h of treatment, the cells were analysed for NF-kB-mediated inflammatory response mediators using the human NF-kB pathway PCR array. It was observed that withaferin A inhibited the expression of NF-kB subunit 2 (NF-kB2) and RELA transcription factors, which play an essential role in the expression of inflammatory chemokines and cytokines. There was also an upregulation of IKBKB and IKBKG (the depletion of these proteins activates NF-kB) and a downregulation of JUN and STAT genes (Figure 5). A downregulation of interleukin-1*β* (IL-1*β*), known for playing an important role in NF-kB-mediated neuroinflammation, was also observed [65]. Pandey et al. (2018) [31] have shown that withanone (WS-2), a compound isolated from the root extract of *W. somnifera*, significantly prevents cognitive impairment and attenuates inflammation and oxidative stress in the intracerebroventricular (ICV) infusion of streptozotocin (STZ) model in rats [31]. WS-2 was tested at doses of 5, 10, and 20 mg/kg, with the most effective dose being 20 mg/kg, after which there was no further increase in activity, and a plateau state was reached. The daily oral administration of WS-2 reduced elevated levels of the pro-inflammatory mediators TNF-*α*, IL-1*β*, IL-6, MCP-1, nitric oxide (NO), lipid peroxidation (LPO), and both *β*- and *γ*-secretase enzymatic activity [31]. WS-2 has shown promise in the treatment of Alzheimer’s disease due to its cognitive benefits and, more importantly, its mechanisms of action related to the basic pathophysiology of the disease, which are not only limited to altering A*β* processing but also include protection against oxidative stress and anti-inflammatory effects.

#### 2.2.4. Effect of *W. somnifera* on Clearance of A*β*

Sehgal et al. (2012) [74] reported the effects of a partially purified *W. somnifera* extract consisting of 75% withanolides and 20% withanosides in APP/PS1 transgenic mice, a model of Alzheimer’s disease [74]. The consecutive oral administration to mice of Ashwagandha extract (a single daily oral dose of 1 g/kg body weight) for 30 days reversed behavioural memory deficits in the radial arm task and decreased A*β* levels in the cerebral cortex and hippocampus but increased those in blood plasma [74]. Low-density lipoprotein receptor-related protein (LRP) in the liver and the soluble form of LRP (sLRP) in the plasma were also upregulated by administering Ashwagandha extract to mice. LRP mediates the efflux of A*β* from the brain to the periphery [75,76]. LRP is cleaved in the liver and released into the blood where it remains as sLRP. The liver-specific knockdown of LRP blocked both sLRP and A*β* accumulation in plasma and A*β* reduction in the brain after extract administration [74]. Therefore, Ashwagandha extract promotes A*β* clearance in the brain via the upregulation of liver LRP. As *W. somnifera* is composed of multiple compounds, it is possible that pathways other than those investigated in this study may contribute to the ultimate therapeutic effect seen. Nevertheless, the potent effect of *W. somnifera* in rapidly clearing A*β* is mainly related to its action in the periphery via increasing levels of liver LRP and sLRP, suggesting that targeting the periphery for A*β* clearance may provide a unique mechanism for the rapid clearance of A*β*, ultimately leading to the reversal of behavioural deficits in AD transgenic mice.

#### 2.2.5. Effect of *W. somnifera* on Scopolamine Toxicity

Scopolamine is a non-selective muscarinic receptor antagonist that blocks cholinergic neurotransmission and impairs learning and memory [77]. The modulation of the cholinergic cascade has been implicated as a primary determinant of the memory and cognitive deficits that characterize several neurodegenerative diseases, including Alzheimer’s disease. In light of this information, scopolamine-induced amnesia was used as a model for age-associated and pathological cognitive dysfunction observed in dementia. As expected, when administered to healthy humans, scopolamine mimicked memory and several other cognitive impairments seen in ageing and dementia. Such compelling evidence led to the use of scopolamine as a standard/reference drug to induce amnesia in healthy humans and animals, as well as for screening therapeutic targets [78]. Konar et al. (2011) treated neuronal and glial cell lines IMR32 and C6 with scopolamine, resulting in the downregulation of brain-derived neurotrophic factor (BDNF) and glial fibrillary acidic protein (GFAP) (both important for normal neuronal and glial cell growth and upregulation of oxidative stress ROS markers) and the downregulation of neuronal cell markers NF-H, MAP2, PSD-95, and GAP-43 and the upregulation of oxidative stress ROS markers. The alcoholic extract of *W. somnifera* leaves or its purified component withanone upregulated the production of NF-H, MAP2, PSD-95, and GAP-43 and downregulated oxidative stress ROS markers [79]. Similarly, *in vivo*, Swiss albino mice treated with an alcoholic extract of *W. somnifera* leaves (i-Extract) showed an increase in brain-derived neurotrophic factor (BDNF) and glial fibrillary acidic protein (GFAP), both of which are downregulated when treated with intraperitoneally administered scopolamine hydrobromide [79,80]. There was also a significant decrease in acetylcholine neurotransmitter levels due to scopolamine injection, as evidenced by the increased enzymatic assay of acetylcholinesterase (AChE) activity and decreased choline acetyltransferase (ChAT) activity, an enzyme that synthesises acetylcholine [80]. With the post- and pre-treatment of i-Extract, a significant decrease in AChE activity and an increase in ChAT activity is found, as well as a significant downregulation of AchE mRNA and a significant upregulation of ChAT mRNA [80]. The i-Extract of Ashwagandha reverses memory loss in scopolamine (SC)-induced mice by primarily targeting the acetylcholine receptors of the muscarinic subtype, which regulate memory processes [81]. Earlier, Bhatnagar et al. (2012) observed direct inhibition of AChE by *W. somnifera* fresh leaf extract when adult albino mice received a single daily dose (4 mL/kg bw) of *W. somnifera* leaf juice prepared in 0.9% normal saline [82]. *In vivo*, when adult male Wistar rats were treated with scopolamine, they showed a significant increase in AChE activity and a decrease in acetylcholine in all brain regions: cerebral cortex (CC), cerebellum (CB), hippocampus (HC) and pons, except in the hypothalamus (HT). Post-treatment with *W. somnifera* methanol extract (ME) and aqueous extracts (AE) (100, 200, and 300 mg/kg bw orally) and donepezil as reference control (5 mg/kg bw orally) caused the inhibition of AChE activity and the restoration of acetylcholine (ACh) levels in a dose-dependent manner [83,84]. The significant response at 300 mg/kg of ME and AE in scopolamine-treated rats suggests that this dose maintains the homeostasis of the cholinergic system in all of the above brain structures [84].

#### 2.2.6. Effect of *W. somnifera* on Glutamate Toxicity

Glutamate is the principal excitatory neurotransmitter in the vertebrate central nervous system, with up to 40% of all synapses being glutamatergic and found in more than 80% of all neurons. Under normal circumstances, glutamate plays a vital physiological role in synaptic plasticity, learning, memory growth, and differentiation. However, the overactivation of its receptors leads to excitotoxicity, which has been shown to be a major causative mechanism in several neuronal disorders, including Alzheimer’s disease [85]. Excitotoxicity promotes a significant increase in cytosolic Ca^2+^ concentration, which triggers various pathological processes such as ROS generation, oxidative stress, ATP depletion, problems in the electron transport chain (ETC), and the loss of mitochondrial membrane potential, ultimately leading to cell death [86]. It has been reported that Ca^2+^ entry through voltage-gated Ca^2+^ channels does not induce cell death, whereas Ca^2+^ influx through N-methyl-d-aspartate receptors (NMDARs) has been associated with significant Ca^2+^-dependent toxicity [87]. Dar et al. (2018) evaluated withanolide A against glutamate-induced excitotoxicity and found that it was able to rescue the cells from glutamate-induced death. Neuro2a cells were incubated with 10 μM of retinoic acid (RA) for 6 days to induce differentiation. After differentiation, the cells were pre-treated with 2.5, 5, 10, and 20 μM withanolide A and with MK-801 (specific NMDA receptor antagonist) and then exposed to 10 mM glutamate for 2 h [88]. Glutamate-induced toxicity is caused by the overactivation of glutamate receptors, resulting in a huge influx of Ca^2+^ ions. This Ca^2+^ accumulation leads to the generation of ROS, increased levels of pro-apoptotic and decreased levels of anti-apoptotic proteins, activation of MAPK family proteins (JNK, ERK1/2, and p38), and inhibition of the PI3K/Akt pathway. However, withanolide A treatment rescued the cells by reducing intracellular calcium levels, attenuating excessive ROS generation, correcting the mitochondrial machinery, and inducing PI3K/Akt activation and the inhibition of MAPK family proteins [88]. The same results were obtained after treatment with withanone [86], one of the active constituents of *W. somnifera*, against NMDA-induced excitotoxicity in retinoic acid, differentiated Neuro2a cells. The cells were pre-treated with 5, 10, and 20 μM doses of withanone and then exposed to 3 mM NMDA for 1 h [86]. MK801, a specific NMDA receptor antagonist, was used as a positive control. The results showed that NMDA induced significant cell death through the accumulation of intracellular Ca^2+^, generation of reactive oxygen species (ROS), loss of mitochondrial membrane potential, crash of Bax/Bcl-2 ratio, release of cytochrome c (cyt c), increase in caspase expression, and induction of lipid peroxidation. All these parameters were attenuated by withanone pretreatment at different doses [86]. These results suggest that withanone may serve as a potential neuroprotective agent. Kataria et al. (2012) investigated the effect of Ashwagandha water extract leaves (ASH-WEX) against glutamate-induced toxicity in retinoic acid-differentiated C6 glioma and IMR-32 cell lines. Glutamate-exposed differentiated cells exhibited massive cell death and neuronal network damage, which was associated with the upregulation of the stress-induced chaperone Hsp70. Pretreatment with ASH-WEX extract (0.1%) significantly inhibited glutamate-induced cell death and HSP70 expression [89]. Furthermore, the analysis of the neuronal plasticity marker NCAM (neural cell adhesion molecule) and its polysialylated form, PSA-NCAM, revealed that ASH-WEX has therapeutic potential for the prevention of neurodegeneration associated with glutamate-induced excitotoxicity [89].

### 2.3. Withania somnifera L. and Parkinson’s Disease

Parkinson’s disease (PD) is the second most common neurodegenerative condition after AD. PD is characterized by the degeneration of dopaminergic neurons of the substantia nigra pars compacta, located in the midbrain and associated with Lewy bodies, which are cytoplasmic inclusions containing insoluble aggregates of alpha-synuclein (a protein present in dopaminergic terminals and important for synaptic vesicle trafficking) that project to the basal ganglia and striatum [90]. The clinical diagnosis of PD is primarily based on motor features such as slowly progressive asymmetric resting tremor, cogwheel rigidity, and bradykinesia, although non-motor features such as anosmia, constipation, depression, and REM sleep behaviour disorder may develop years before motor deficits [90]. There are many theories about the causes of PD, and although they differ, they all involve *α*-synuclein aggregation. Some theories suggest that PD is caused by mitochondrial dysfunction, oxidative stress, low dopamine levels, and microglial dysfunction [91,92]. The only treatments available for PD are symptomatic; the most common are L-DOPA or dopamine agonist treatments.

#### 2.3.1. Effect of *W. somnifera* on Oxidative Stress

Impaired antioxidant defence mechanisms and the increased generation of oxidative free radicals have been implicated in neurodegenerative diseases such as PD. Superoxide dismutase (SOD), catalase (CAT), and glutathione peroxidase (GPX) are the main free radical-scavenging enzymes. The dysfunction of these enzymes leads to the accumulation of toxic free radicals and consequent degenerative progression of the disease [93]. The active glycowithanolides of *W. somnifera* have been found to increase cortical and striatal concentrations of the antioxidant enzymes SOD, CAT, and GPX [94]. Jayawanth Manjunath (2013) investigated the neuroameliorative effects of *W. somnifera* in prepubertal male mice treated with rotenone (ROT), a commonly used natural pesticide. ROT is a classical high-affinity specific inhibitor of mitochondrial complex I and is capable of causing mitochondrial dysfunction and oxidative stress in specific brain regions, namely the cerebellum (CB) and striatum (ST) of PP mice, mimicking PD. In this study, groups of mice (n = 6) were orally administered (by gavage) *W. somnifera* root powder (carried in normal saline) at three doses, viz. 100, 200, and 400 mg/kg bw/d, for 4 weeks [95]. Oral supplementation with *W. somnifera* significantly reduced endogenous oxidative markers and nitric oxide levels in ST and CB brain regions, suggesting its potential to attenuate neurotoxic damage. Furthermore, the increased levels of GSH and thiols in these brain regions with a concomitant increase in the activity levels of antioxidant enzymes (such as SOD and GPx) may render the brain regions less susceptible to neurotoxic-mediated oxidative dysfunction [95]. Manjunath and Murlidhara (2015) investigated the neuroameliorative effects of *W. somnifera* root extract standard powder, WSP (withanolides, 2.57%; withaferin A, 2.38%), in a rotenone (ROT) model of *Drosophila melanogaster* (Oregon-K). *W. somnifera* conferred significant protection against ROT-induced lethality, and surviving flies exhibited an improved locomotor phenotype when exposed to three concentrations of *W. somnifera* standardised powder (0.005, 0.01, 0.05%) [96]. Furthermore, biochemical studies revealed that *W. somnifera* significantly reduced ROT-induced oxidative stress. *W. somnifera* caused a significant increase in the levels of reduced GSH/non-protein thiols. Moreover, the altered activity levels of succinate dehydrogenase, membrane-bound enzymes viz. NADH-cytochrome c reductase and succinate cytochrome c reductase, were significantly restored to normal [96]. The common fruit fly, *Drosophila melanogaster* (Dm), is a simple animal species that has made significant contributions to the development of neurobiology, since leucine-rich repeat kinase 2 (LRRK2) loss-of-function mutants in the WD40 domain provide a very interesting tool for studying the physiopathology of Parkinson’s disease (PD). LRRK2 mutants have a significantly reduced lifespan and impaired motor function and mitochondrial morphology compared to WT flies on a 1% *W. somnifera* enriched diet [97]. 6-Hydroxydopamine (6-OHDA) is one of the most widely used rat models of PD, inducing its toxic manifestations through oxidative stress. The anti-Parkinsonian effect of *W. somnifera* extract was evaluated and reported to have potent antioxidant, anti-peroxidant, and free radical-quenching properties in various disease conditions. Aqueous [98] and ethanolic [99] *W. somnifera* root extract was found to significantly reverse the levels of reduced glutathione, GPx, SOD, and CAT in a dose-dependent manner as compared to 6-OHDA rat model *in vivo* or *in vitro* on 6-hydroxydopamine (6-OHDA) human neuroblastoma SH-SY5Y cell line [98,99]. Similarly, Prakash et al. (2013), through their work, showed the neuroprotective function of ethanolic extract of *W. somnifera* root against maneb (MB), a fungicide, and paraquat (PQ), an herbicide. MB–PQ induced dopaminergic neurodegeneration in a male Swiss albino mouse model of PD. According to their work, *W. somnifera* extract is able to inhibit the oxidative stress occurring in nigrostriatal tissues while increasing the number of tyrosine hydroxylase-positive cells in the SN region of the brain of MB–PQ-induced PD mice and also improving motor coordination [100,101]. MPTP (1-methyl-4-phenyl-1,2,3,6-tetrahydropyridine) is another neurotoxic agent that has been used to induce PD in animal models by inhibiting the mitochondrial electron transport chain. In the brain, MPTP is metabolized by monoamine oxidase (MAO) to its active toxin, MPP^+^. This enzymatic conversion of MPTP to MPP^+^ has been shown to be associated with generating free radicals [102]. The administration of MPTP to male Swiss albino mice at a concentration of 20 mg/kg bw/day injected intraperitoneally for 4 consecutive days resulted in the increased production of reactive oxygen species (ROS) and increased levels of SOD, CAT and malondialdehyde (MDA) activities in both the midbrain and striatum compared to the control brain (*p* < 0.05), while the levels of GPx and GSH activities were reduced in both the midbrain and striatum compared to the respective control (*p* < 0.05) [103,104,105,106]. The oral treatment of PD mice with *W. somnifera* root extract [103,104,105] or leaf extract [106] reduced the elevated levels of MDA, CAT, and SOD observed in the brain of PD model mice. There was a remarkable improvement in the levels of GSH and GPx in the PD mouse fed with *W. somnifera* leaf or root extracts [103,104,105,106]. Mice treated with MPTP also show physiological abnormalities similar to those seen in Parkinson’s patients, as determined by the Hang and Rotarod tests. This is because substantia nigra (SN) neurons are more susceptible to oxidative stress due to the presence of high levels of iron and low levels of endogenous antioxidants compared to other brain regions [107]. The oral treatment of PD mice with *W. somnifera* root extract showed improved motor function and all physiological parameters [103,105,106]. Thus, *W. somnifera* has strong antioxidant potential, and its ROS scavenging property plays an important role in preventing PD by counteracting neurodegeneration.

#### 2.3.2. Effect of *W. somnifera* on Catecholamines Level

The neurotransmitter dopamine (DA) plays a key role in motor control and body movement. Oxidative stress and reduced levels of catecholamines are the contributing factors to neurodegeneration in PD [108], leading to the loss of motor function in PD patients [109,110]. An animal model of PD has been created by administering MPTP (1-methyl-4-phenyl-1,2,3,6-tetrahydropyridine) in doses of 20 mg/kg body weight/day for 4 days (i.p.). The administration of MPTP inhibits mitochondrial complex I, leading to increased reactive oxygen species (ROS) production. Rajasankar et al. (2009) analysed catecholamines such as dopamine (DA), 3,4-dihydroxyphenylacetic acid (DOPAC), and homovanillic acid (HVA) in the striatum of *W. somnifera*-treated and untreated PD mice; the oral treatment of PD mice with *W. somnifera* root extract (100 mg/kg body weight) for 7 days or 28 days increased DA, DOPAC, and HVA levels in the corpus striatum (dopamine and its metabolites, DOPAC and HVA, were analysed using reversed-phase ion-pair chromatography combined with electrochemical detection under isocratic conditions) [105] and also reduced motor impairment compared to the control group [105,106]. Thus, their work inferred the medicinal benefits of *W. somnifera*, which increases the levels of catecholamines and antioxidants and prevents lipid peroxidation in the corpus striatum of PD mice. Prakash et al. (2014) studied the effect of ethanolic root extract *W. somnifera* on dopamine and its metabolites in the SN region of MB–PQ-induced mouse models of PD. A reduction in dopamine and its metabolites was found in the brains of PD mice compared to controls. Furthermore, treatment with the ethanolic root extract of *W. somnifera* significantly improved dopamine, DOPAC, and HVA levels (estimated using a standard HPLC method) compared to untreated PD mice [101]. It is, therefore, possible to state that *W. somnifera* has the ability to increase catecholamine levels and combat PD-like disorders.

#### 2.3.3. Effect of *W. somnifera* on Apoptotic Pathways

Apoptosis, or programmed cell death, is a tightly regulated process that leads to the active suicide of cells under certain circumstances. The dysregulation of programmed cell death has been implicated as the major cause of neurodegenerative disease [111]. Bcl-2 is an anti-apoptotic protein that suppresses cell death by inhibiting the action of a pro-apoptotic protein, Bax. Thus, the ratio of Bcl-2 to Bax determines whether a cell will survive or succumb to apoptosis. Interestingly, one study has suggested that the overexpression of Bcl-2 helps to attenuate MPTP-induced neuronal cell death [112]. Prakash et al. (2014) showed that Bcl-2 expression was significantly downregulated, while Bax expression was significantly upregulated in male Swiss albino mice (weighing 25 ± 5 g) treated with a combination of PQ and MB. Furthermore, it was observed that the ethanolic root extract of *W. somnifera* increased the level of anti-apoptotic (Bcl-2) proteins and decreased the level of pro-apoptotic (Bax) proteins in the MB–PQ model of PD [101]. *W. somnifera* has, thus, been shown to be able to regulate the levels of the apoptotic proteins Bcl-2 and Bax. It can be hypothesized that *W. somnifera* has the ability to overcome neurological disorders such as PD by regulating the apoptotic pathway.

#### 2.3.4. Effect of *W. somnifera* on Aggregation of α-Synuclein Protein

The most common pathological changes observed in patients with PD are the degeneration of dopaminergic neurons, particularly in the substantia nigra region of the midbrain, the aggregation of *α*-synuclein in neurons, and decreased brain dopamine levels. The effect of the extract of *W. somnifera* in reducing the aggregation of *α*-synuclein protein was evaluated *in vivo* using a *Caenorhabditis elegans* PD model. A strain of *Caenorhabditis elegans* (NL5901) with a muscular expression of *α*-synuclein labelled with a yellow fluorescent protein (YFP) was used to analyse the effect of *W. somnifera* methanolic extract [113] and withanolide A [114] on the aggregation of protein α-synuclein. The fluorescence intensity at the worms’ anterior end was used to measure α-synuclein aggregation. NL5901 worms treated with *W. somnifera* extracts showed significantly lower *α*-synuclein fluorescence intensity than untreated NL5901 worms, suggesting a beneficial effect of *W. somnifera* for PD patients. Previously, *W. somnifera* extract was reported to inhibit A*β* protein aggregation through enhanced protein clearance [74]. Thus, *W. somnifera* may act on multiple cellular pathways, including protein aggregation clearance pathways.

#### 2.3.5. Synergistic Effect of *W. somnifera*

Gupta and Avtar (2009) worked on the synergistic effect of *W. somnifera* and L-DOPA in inhibiting haloperidol-induced catalepsy in mice. It was demonstrated that the anti-cataleptic effect of *W. somnifera* could be attributed to its polyphenols, which are responsible for the direct scavenging of free radicals, and the inhibition of lipid peroxidation in the central nervous system [115]. *W. somnifera* and *Mucuna pruriens* (L.) DC. are traditional herbal plants known to have neuroprotective effects due to the presence of L-DOPA in *M. pruriens* seed powder and withanoloides in *W. somnifera* root extract [116]. Therefore, the synergistic effect of *W. somnifera* and *M. pruriens* in PD mice induced by chronic exposure to 1-methyl-4-phenyl-1,2,3,6-tetrahydropyridine (MPTP) [117] and paraquat (PQ) [116] was investigated, and all neurochemical variables, oxidative stress, and physiological abnormalities were found to be significantly improved compared to the brain of untreated PD mice. According to Prakash et al. (2013a), exposure to PQ increases nitrite levels in the nigrostriatal region. Therefore, this investigation found that *M. pruriens* and *W. somnifera* co-exposure ameliorated the nitrite levels in PQ-treated mice. This decrease in nitrite levels by *M. pruriens* and *W. somnifera* could be attributed to the antioxidant properties of *M. pruriens* [118] and *W. somnifera* [11] plant extracts. Malondialdehyde (MDA), a lipid peroxidation product, has also been used as a marker of oxidative damage [100,116] showing that after the treatment of mice with PQ, MDA levels were highly elevated compared to controls. However, MDA levels were significantly improved after co-treatment with *M. pruriens* and *W. somnifera*. Thus, the combined treatment showed a significant effect compared to *M. pruriens* and *W. somnifera* treatment alone. Recently, Vegh et al. (2021) evaluated the neuroprotective efficacy of the oral administration of ethanolic Ashwagandha root extract (ASH) and a water-soluble formulation of coenzyme Q10 (Ubisol-Q10) alone and in combination in a paraquat-induced PD rat model. The combined treatment resulted in better preserved neuronal morphology than Ubisol-Q10 or ASH alone. The combination treatment enhanced the activation of pro-survival astroglia and inhibited pro-inflammatory microglia. While antioxidant effects were seen with both agents, Ubisol-Q10 activated autophagy, while Ashwagandha showed a better anti-inflammatory response. Thus, the combined treatment resulted in oxidative stress and microglia inflammation inhibition and autophagy and pro-survival astroglia activation [119]. Therefore, the pioneering work on the synergistic effects of *W. somnifera* with *M. pruriens*, *W. somnifera* with L-Dopa, and *W. somnifera* with Ubisol-Q supported the efficacy of *W. somnifera* in treating PD.

### 2.4. Withania somnifera L. and Huntington’s Disease

Huntington’s disease (HD) is an autosomal dominant neurodegenerative disease characterized by progressive motor dysfunction such as chorea and dystonia and behavioural and cognitive decline. The disease is caused by an expansion of a cytosine-adenine-guanine (CAG) trinucleotide, which is repeated in the coding region of the HD gene located on the short arm of chromosome 4. This section encodes the protein huntingtin (HTT), which is widely expressed and has multiple functions in human neurons [120]. As the number of CAG repeats increases (>35 CAG repetitions), HTT becomes susceptible to misfolding, thereby forming insoluble aggregates in the nucleus and cytoplasm of neurons [121]. These aggregates accumulate and lead to cell dysfunction and apoptosis, eventually causing the severe atrophy of the affected brain areas [122]. Tetrabenazine (TBZ) is the only drug currently approved by the FDA to treat chorea in HD. TBZ reversibly inhibits the vesicular monoamine transporter 2 (VMAT-2) in the central nervous system. Since VMAT-2 packages serotonin, dopamine, and norepinephrine from the cytoplasm into presynaptic vesicles, its inhibition leads to premature degradation of these monoamines. Since dopamine is necessary for fine motor movement, its transmission inhibition leads to a reduction in hyperkinetic movements [123].

The compound 3-nitropropionic acid (3-NP) is a potent neurotoxin; it induces oxidative and nitrosative stress, inhibits complex II of the mitochondrial electron transport chain (resulting in a deficit of ATP), and is responsible for biochemical and neurobehavioural changes comparable to those observed in HD. In an animal model, symptoms of HD were artificially induced by the intraperitoneal administration of 3-NP. The intraperitoneal administration of 3-NP to male Wistar rats caused a loss of body weight and a decline in motor function (locomotor activity and impaired rotarod activity) [124,125]. Chronic treatment with *W. somnifera* root extracts improved 3-NP-induced behavioural, biochemical, and enzymatic changes (*p* < 0.05) [125]. Biochemical analysis revealed that the systemic administration of 3-NP significantly increased lipid peroxidation and nitrite and lactate dehydrogenase enzyme levels, depleted antioxidant enzyme (superoxide dismutase and catalase) levels, and blocked ATP synthesis by inhibiting mitochondrial complex activity in different regions (striatum and cortex) of the brain [125]. These findings suggest that the neuroprotective effects of *W. somnifera* are mediated via its antioxidant properties. The involvement of the GABAergic system in the pathogenesis of HD has also been reported. Ashwagandha acts through the GABAergic system and its antioxidant potential restores acetylcholinesterase and glutathione enzyme levels and improves cognitive function, making it a suitable candidate for treating HD [124]. Another study in mice demonstrated the beneficial effects of withaferin A isolated from Ashwagandha. The inability of cells to maintain proteostasis is a sign of ageing and a hallmark of many neurodegenerative diseases, including HD. *In vitro* and *in vivo*, treatment with low doses of withaferin A in the R6/2 transgenic mouse model of HD and HD150Q cells has been shown to strongly activate the heat shock response (HSR) through the activation of heat shock factor 1 (HSF1) by thiol oxidation to ameliorate impaired proteostasis and delay disease progression [126]. At higher doses, withaferin A has been shown to inhibit proteasomal dysfunction and the induction of autophagy, and these effects may be related to its effect on thiol modification in the cell. HD mice treated with withaferin A lived significantly longer, and behavioural and motor deficits were reversed, including a reduction in body weight [126]. Biochemical studies confirmed the activation of heat shock, reduction of mutant huntingtin aggregates, and improvement of striatal function in the brain of mice. In addition, withaferin A significantly reduced inflammatory processes, as indicated by reduced microglial activity [126]. *W. somnifera* root extract and its constituent withanolide A have also been shown to significantly improve cognitive behaviour (as measured by the Morris water maze and elevated plus maze tests) and motor activity (impairment of muscle activity as measured by the rotarod and limb withdrawal tests) [124,127]. This improvement has been attributed to inhibiting oxidative stress, restoring antioxidant status, and enhancing acetylcholinesterase enzyme activity with *W. somnifera* supplementation [124,125]. Thus, *W. somnifera* extracts or their purified component withaferin A may have potential therapeutic benefits in HD.

### 2.5. Withania somnifera L. and Amyotrophic Lateral Sclerosis Disease (ALS)

ALS differs from AD, PD, and HD primarily because it is a motor neuron disease with pathology in the spinal cord and brain. Although 90–95% of cases are sporadic, the familial form can account for 5–10% of the total number of patients, with many genes involved. The most common mutations in both familial and sporadic cases are copper–zinc superoxide dismutase (SOD1 mutations leading to free radical toxicity, cascading inflammatory responses, and excessive concentrations of glutamate), TAR-DNA binding protein 43 (TDP-43), fused in sarcoma (FUS) and C9ORF72, with an expansion of the GGGGCC hexanucleotide. Although the causes of ALS vary, a common histopathological sign is the presence of intracytoplasmic inclusions such as Bunina bodies, but also SOD1, ubiquitylated, and TDP-43 inclusions [128]. As a result, many of the beneficial properties of *W. somnifera* extracts or their constituents observed in AD and PD may not be sufficient to explain their neuroprotective properties in ALS. Promoting neurite and/or dendrite outgrowth may be beneficial to some extent in alleviating ALS-associated frontotemporal lobar degeneration (FTD), and their anti-cholinesterase property may be beneficial in the presymptomatic state of familial ALS (fALS) [129]. Therefore, the effect of *W. somnifera* in relation to ALS must be specific to this disease. TDP-43 protein inclusions in spinal motor neurons are hallmarks of ALS and/or FTD. TDP-43 has been shown to bind to and act as a co-activator of the p65 subunit of nuclear factor kappa B (NF-κB) in ALS patient samples and in a transgenic mouse model generated using genomic fragments encoding wild-type (WT) human TDP-43 [130]. A four-fold increase in p65 mRNA levels was observed in spinal cords from sporadic ALS cases, and p65 was found to be predominantly localized in the neuronal nucleus. The interaction between TDP-43 and p65 was not observed in either non-ALS control spinal cords or non-transgenic animals, leading to the proposal that the interaction between these two proteins is involved in the pathogenic process of the disease [130]. To validate the above findings, withaferin A was used as an NF-κB inhibitor, and interestingly, when administered to transgenic mice overexpressing WT human TDP-43, the disease pathology was found to be alleviated [130]. A marked reduction in the glial activation pattern was observed in the spinal cord of withaferin A treated animals, as visualized by the reduced expression of Mac-2 and GFAP and a reduction in microglial COX-2 levels [130]. In addition, withaferin A treatment improved motor performance in transgenic mice expressing human WT TDP-43 and enhanced the innervation of neuromuscular junctions (NMJs) [130]. Similarly, the withaferin A treatment of transgenic mice expressing human mutant TDP-43 (G348C) was found to ameliorate motor deficits in the rotarod test compared to TDP-43 (G348C) mice [130]. In cultured primary neurons overexpressing WT or mutant TDP-43, NF-κB inhibition by withaferin A reduced their susceptibility to glutamate toxicity [130]. Hence, targeting the p65 pathway appeared to be a potential therapeutic approach for ALS. To further investigate the role of withaferin A in ALS, the drug was next administered to two other mouse models of fALS—SOD1 (G93A) and SOD1 (G37R) [131,132]. The SOD1 (G93A) is an aggressive model of the disease, but withaferin A treatment resulted in a modest increase in median survival of 8–9 days. In the SOD1 (G37R) model, which is comparatively less acute, survival was extended by approximately 18 days [131]. In addition to delaying the rate of disease progression, this prolongation of survival could be attributed to a significant reduction in misfolded SOD1 species in the spinal cord of withaferin A treated animals. This reduction in SOD1 misfolding by withaferin A treatment may be related in part to increased levels of heat shock proteins (HSPs). Withaferin A is a known proteasome inhibitor, and this, in turn, could induce endoplasmic reticulum (ER) stress response [133] and also induce autophagy [134]. An important role of the ER stress response is the recruitment of chaperones to correct protein misfolding. In withaferin A treated mice, HSP70 and HSP25 (equivalent to human HSP27) were found to be significantly increased, which could explain normal protein function and overall neuroprotection. However, the protective effect of withaferin A was only observed when administered at an early stage of the disease and not when drug treatment was initiated at a late stage [131]. Recently, Dutta et al. (2018) tested a whole root extract of *W. somnifera* administered orally to pre-symptomatic SOD (G93A) mice from 50 days of age. Although the *W. somnifera* extract did not significantly alter the survival of SOD (G93A) mice, it did confer neuroprotection and improve motor performance [132]. The effect of the oral administration of *W. somnifera* extract was also tested in transgenic mice expressing the human mutant hTDP43 (A315T), a mouse model of motor and cognitive dysfunction with abnormal cytoplasmic TDP-43 accumulation [135]. The results showed that treatment with *W. somnifera* was able to improve motor performance in the rotarod test and cognitive function in the passive avoidance test in hTDP43 (A315T) transgenic mice. Remarkably, the microscopic examination of CNS samples revealed that the treatment of hTDP43 (A315T) mice with *W. somnifera* extract resulted in the translocation of mislocalised cytoplasmic hTDP43 to the nucleus of spinal motor neurons [136]. Withaferin A has also been reported to bind to the astrocyte intermediate filament proteins vimentin and glial acidic fibrillary protein (GFAP). Both proteins are known to be elevated in ALS [130,131]. In cultured astrocytes, withaferin A binding to these two intermediate filament proteins downregulates soluble vimentin and GFAP expression to induce G0/G1 cell cycle arrest, which may lead to a reduction in gliosis-dependent inflammation. Withaferin A can also bind to peripherin, another intermediate filament protein, which has been shown to be involved in neurite outgrowth during development and axonal regeneration but may also be responsible for protein aggregation and motor neuron death in ALS [137,138]. In Veh-treated transgenic mice, peripherin was found to be prominently expressed. However, in the spinal cord of hTDP43 (A315T) mice treated with ASH, there was a marked reduction in peripherin levels [136]. It can, therefore, be concluded that either *W. somnifera* extracts or their purified components may have potential therapeutic benefits in ALS.

### 2.6. Clinical Studies

*In vivo* investigations have shown that supplementation with *W. somnifera* can promote cognitive function and improve memory. These activities have also been tested in humans; there is evidence that *W. somnifera* supplementation (500–600 mg/day for 8–12 weeks) improved memory, executive function, sustained attention, and processing speed in people with early dementia [139]. In a prospective, randomised, double-blind, placebo-controlled pilot study, 50 subjects over 35 years of age with mild cognitive impairment (MCI) were treated with either Ashwagandha root extract (KSM-66 Ashwagandha, which is a 100% aqueous extract of *W. somnifera* roots containing 5% withanolides, as determined by high performance liquid chromatography) at a concentration of 300 mg twice daily or placebo capsules (containing starch as an inert filler) for eight weeks. All subjects were assessed at baseline, four weeks, and eight weeks; the primary outcome measure for this study was a battery of cognitive tests to assess memory, visuospatial, executive function, and attention. Daily treatment with Ashwagandha for eight weeks produced significant improvements over placebo on a battery of cognitive tests designed to assess memory (*p* ≤ 0.05), executive function (*p* ≤ 0.03), and attention and information processing speed (*p* < 0.01). Executive function was assessed using the Wisconsin Card Sort Test and the Eriksen Flanker task. Ashwagandha treatment was associated with improved executive function parameters measured by these instruments, as indicated by a significant increase in these scores compared to placebo at eight weeks (*p* < 0.05). Ashwagandha treatment was also associated with significant improvements in attention and information processing speed compared to placebo, as indicated by scores on the Trail Making Test Part A and the Mackworth Clock Test at baseline and eight weeks (*p* < 0.05) [139]. These studies support the role of Ashwagandha in enhancing memory and improving executive function in people with MCI, which is often a precursor to Alzheimer’s disease.

The effect of *W. somnifera* extract on cognitive and psychomotor performance in healthy human participants has also been investigated [140]. In this prospective, double-blind, multiple-dose, placebo-controlled, crossover study, 20 healthy male participants aged 20–35 years were randomised to receive two capsules of 250 mg each (each capsule of *W. somnifera* extract contains not less than 10% withanolide glycosides, not more than 0.5% withaferin-A, and not less than 32% oligosaccharides) twice daily (i.e., morning and evening) (1000 mg/day) of an encapsulated dried aqueous extract of the roots and leaves of *W. somnifera* or a matching placebo (identical in appearance and containing 49.7% (*w*/*w*) microcrystalline cellulose, 49.5% (*w*/*w*) lactose and 0.69% (*w*/*w*) magnesium stearate) for 14 days. Cognitive and psychomotor performance were assessed pre-dose (day 1) and 3 h post-dose on day 15 using a battery of computerised psychometric tests. It was observed that *W. somnifera* extract significantly reduced reaction time (RT) in Simple reaction test, Choice discrimination test, Digit symbol substitution test, Digit vigilance task, and Card sorting test after 14 days of treatment (1000 mg/day) compared to placebo, indicating its positive effect on psychomotor function.

Recently, another clinical trial investigated the effects of Ashwagandha on cognition in 13 healthy adults [141] instructed to ingest 400 mg of a placebo (PLA) or Ashwagandha root and leaf extract (ASH) in a double-blind, placebo-controlled, crossover study. Participants underwent cognitive function tests every hour for 6 h, demonstrating that participants’ ability to stay alert was better maintained with ASH, helping to prevent mental fatigue. The present findings provide the first clinical evidence that supplementation with Ashwagandha in healthy human subjects improves cognitive performance, including improving sustained attention and enhancing short-term/working memory even at low dosage (400 mg) [141]. 

These investigations corroborate results from animal studies showing that *W. somnifera* is able to improve memory and enhance cognitive function by modulating cholinergic neurotransmission. Extracts of *W. somnifera*, indeed, produce an increase in cortical muscarinic acetylcholine capacity, which may partly explain the cognition-enhancing effects demonstrated in *in vivo* and clinical investigations. However, it must be considered that most of the clinical studies analysed assessed cognitive function in healthy subjects, indicating a potential risk of bias in future application in patients with neurodegenerative diseases.

## 3. Materials and Methods

### 3.1. Search Strategy

The systematic search of the literature was performed based on the Preferred Reporting Items for Systematic Reviews and Meta-Analyses (PRISMA) guidelines [36] and included all articles published until January 2024. The items considered were found in two search databases: Scopus and PubMed. For the search, the species *Withania somnifera* was paired with the following words: “neurodegenerative diseases”, “neuroprotective effects”, “Huntington”, “Parkinson”, “Alzheimer”, “Amyotrophic Lateral Sclerosis”, and “neurological disorders”. Only English-language publications were considered.

### 3.2. Study Selection

The criteria for selecting studies for the review included pre-clinical research (both *in vivo* and *in vitro*) as well as clinical studies involving *Withania somnifera* (L.) Dunal and its related active metabolites Only English articles and those containing the keywords in the title or abstract were considered. Other review articles, editorials, letters, manuscripts, meta-analyses, conference papers, short surveys, and book chapters were not considered. All selected articles were screened closely to exclude or include manuscripts not complying with the specified criteria. Two investigators (M.P. and V.L.) screened titles, abstracts, and full texts to select manuscripts. In cases of disagreement, additional independent reviewers (L.M. and N.T.T.) were consulted. All selected articles were carefully evaluated to determine their adherence to the specified criteria for inclusion or exclusion.

### 3.3. Data Extraction

Two investigators (M.P. and V.L.) independently extracted the data from each selected article, considering information regarding the study design, experimental models, doses used, main results, and general mechanism of action. Studies reporting results for the chosen neurodegenerative diseases from the year 2000 onwards were selected. Articles without relevant results were not considered (Figure 3); in case of discordance, two other authors were consulted to evaluate the possible presence of data extraction discrepancy. To ensure the fluency of the text and understanding of the results, selected articles were grouped by pathologies and summarized, considering the mechanism of action. 

Table 2 summarizes the selected reports’ characteristics, including the tested extract or compound, the model used or the type of assay performed, the treatment time, administrated doses, and demonstrated biological activity.

### 3.4. Methodological Quality Assessment

The quality assessment of each article and the risk of bias was conducted by considering the checklist from the Cochrane Handbook for Systematic Reviews of Interventions, adjusted explicitly for animal intervention studies (SYRCLE’s) [144] and clinical trials [144]. The quality evaluation of each study was performed considering the absence or presence of all information reported in Table 3 and Table 4. Articles that did not meet all criteria were classified as articles with a medium risk of bias, while papers that did not meet these criteria were included in the high risk of bias group. Finally, manuscripts that meet all parameters were assessed as having a low risk of bias.

## 4. Conclusions

Modern conventional medicines suffer from limitations such as increased resistance, unavoidable side effects, loss of efficacy due to prolonged use, and high cost. This has forced researchers to look for bioactive therapeutic compounds and drugs from natural sources such as botanicals. Neuroprotective drugs for clinical use are rare, and most of them, although effective when tested in animal models, have failed to show similar results in clinical trials. Commonly used drugs to treat neurodegenerative diseases include galantamine, donepezil, rivastigmine, and selegiline, but they can only delay disease progression and provide symptomatic relief, showing in some cases, also, side effects. It is, therefore, necessary to focus on the prospective possible use of other herbal and natural medicines like *W. somnifera* that might provide overall protection for mental and neuronal health. *In vitro* and *in vivo* studies, in fact, demonstrated the ability of this species to promote neuronal health by influencing abnormal protein deposition, thus highlighting a potential role in reversing neurodegeneration. Ashwagandha contains a wide range of bioactive compounds, and those that have shown the most promising effectiveness on neurodegenerative diseases are withanolides and their derivatives. 

This systematic review provides the analysis of recent investigations highlighting the pharmacological and neuroprotective potential of *W. somnifera* extracts and compounds. However, most of the revised data are based on *in vitro* and *in vivo* studies, and only a few clinical data are available in the literature regarding the potential role of *W. somnifera* in improving cognitive and psychomotor performance; moreover, most of them are on healthy patients. Therefore, to enhance the knowledge of *W. somnifera’s* effectiveness in preventing or treating neurodegenerative diseases such as AD, PD, ALS, and HD, further studies on human subjects affected by neurodegenerative diseases are required.

## Figures and Tables

**Figure 1 plants-13-00771-f001:**
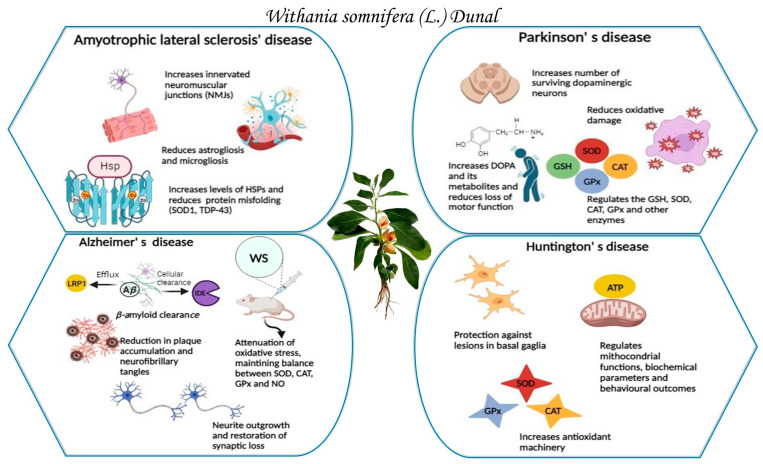
Graphical depiction of neuroprotective effects of *Withania somnifera* (L.) Dunal.

**Figure 2 plants-13-00771-f002:**
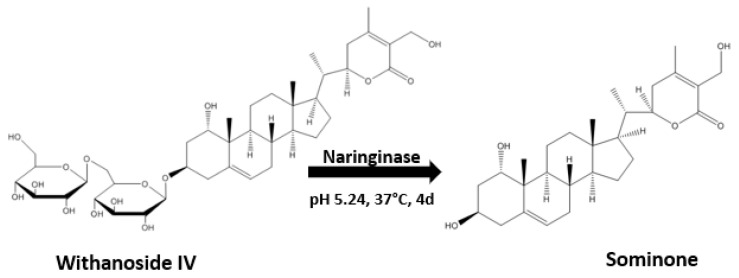
Synthesis of sominone from withanoside IV.

**Figure 3 plants-13-00771-f003:**
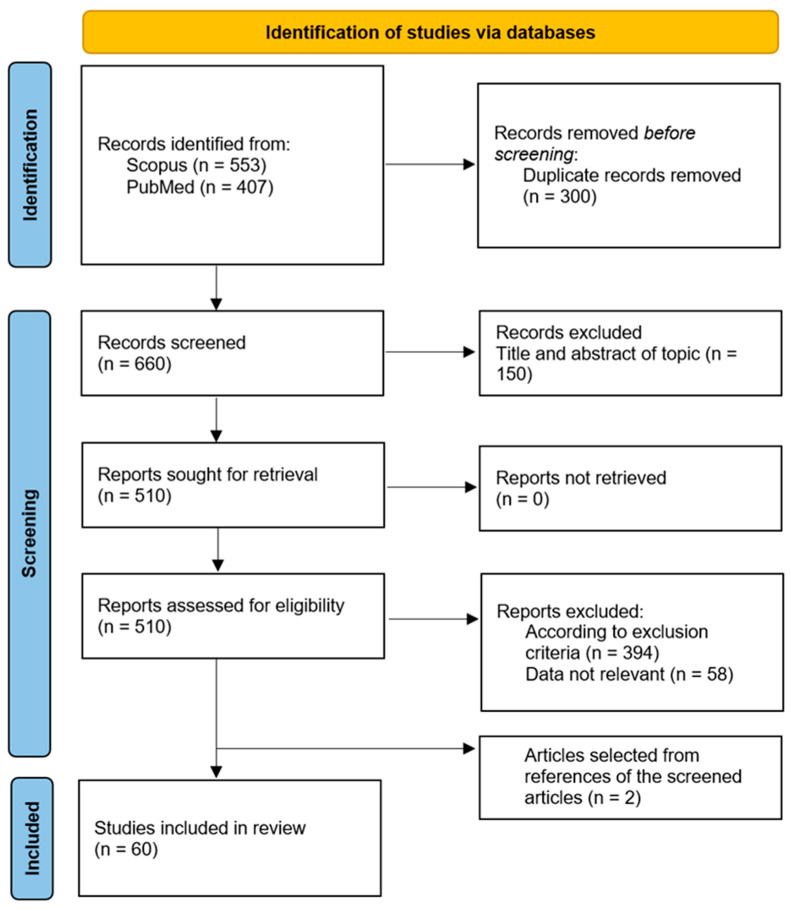
Diagram of systematic review literature search results based on the Preferred Reporting Items for Systematic Reviews and Meta-Analyses (PRISMA) guidelines [36].

**Figure 4 plants-13-00771-f004:**
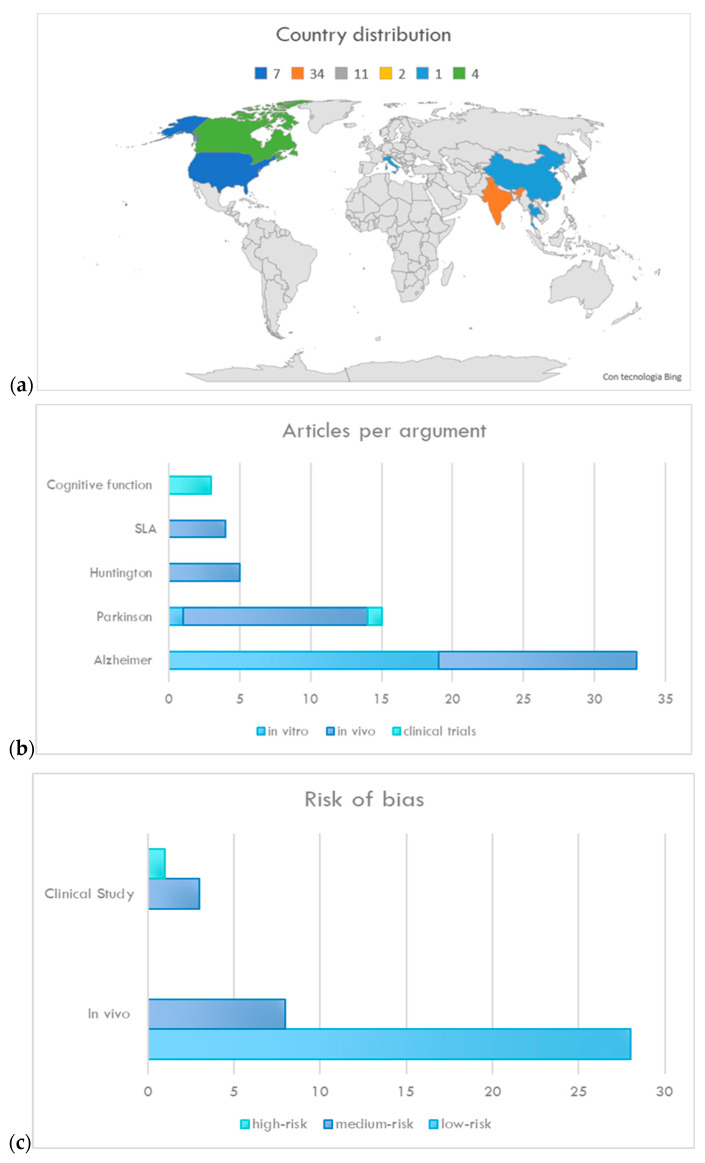
(**a**) Representation of author origin country distribution, (**b**) number of articles per subject area, and (**c**) the quality assessment based on a checklist adapted from the Cochrane Handbook for Systematic Reviews of Interventions. The *in vivo* and clinical studies have been classified as high (green bar), medium (blue bar), and low risk of bias (light blue bar).

**Figure 5 plants-13-00771-f005:**
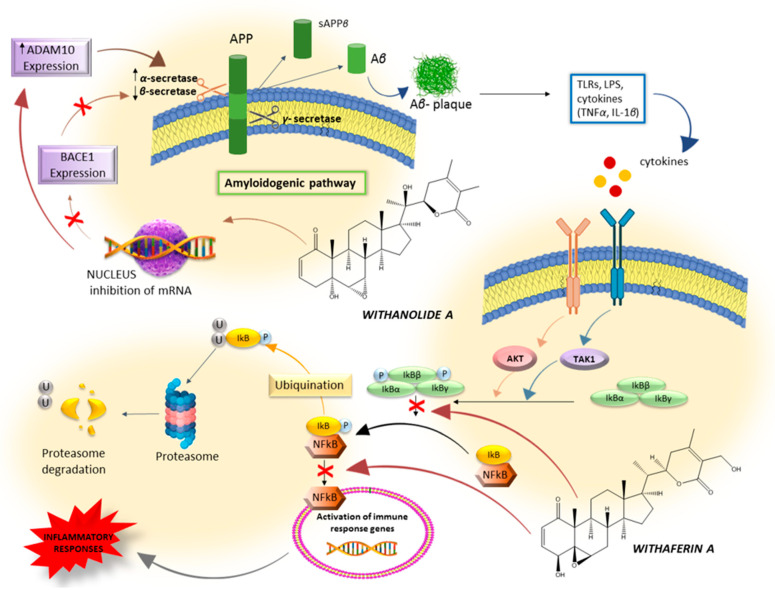
Schematic representation showing the target sites of withanolide A and withaferin A action in the amyloidogenic pathway leading to inflammatory responses. Withanolide A increases a disintegrin and metalloproteinase domain-containing protein 10 (ADAM10) expression and decreases beta-site amyloid precursor protein cleaving enzyme 1 (BACE1) expression, resulting in increased production of soluble and non-toxic APPα; withaferin A inhibits NF-κB signalling.

**Table 1 plants-13-00771-t001:** Main bioactive compounds of *Withania somnifera* (L.) Dunal with neuroprotective activities.

Compound	Molecular Formula	Structure	Part of *W. somnifera* (L.) Dunal	Ref.
**WITHANOLIDE STEROIDAL LACTONES**
Withaferina A	C_28_ H_38_ O_6_	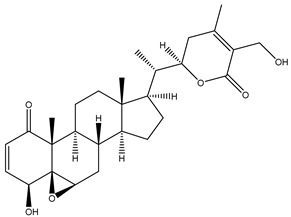	Leaves, Roots	[16]
Withanolide A	C_28_ H_38_ O_6_	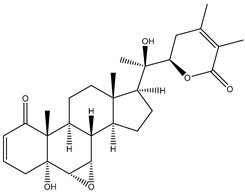	Leaves, Roots	[16]
Withanolide B	C_28_ H_38_ O_5_	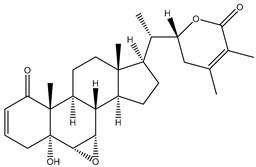	Leaves, Roots	[16]
Withanone	C_28_ H_38_ O_6_	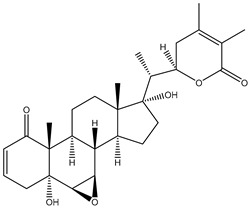	Leaves, Roots	[16]
Sominone (aglycone of withanoside IV)	C_28_ H_42_ O_5_	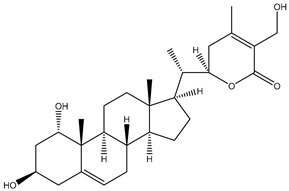	Roots (it was obtained by enzymatic hydrolysis with beta-glucosidase from withanoside IV)	[22]
**WITHANOLIDE GLYCOSIDES**
**WITHANOSIDE**				
Withanoside IV	C_40_ H_62_ O_15_	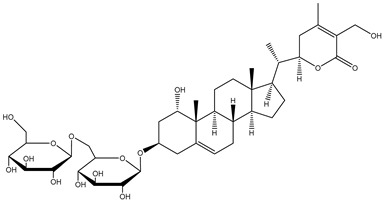	Roots, fruits	[23,24]
Withanoside V	C_40_ H_62_ O_14_	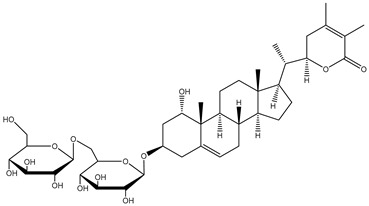	Roots, fruits	[23,24]
Withanoside VI	C_40_ H_62_ O_15_	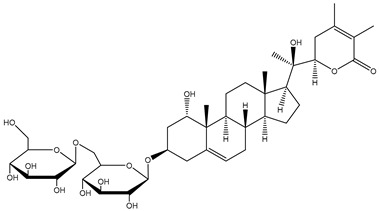	Roots, fruits	[23,24]
WITHANAMIDE(general structure)		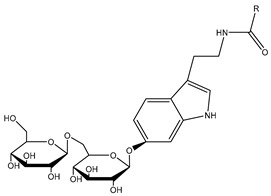		
Withanamide A	C_40_ H_62_ N_2_ O_13_	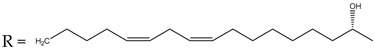	Roots, fruits	[24]
Withanamide C	C_38_ H_63_ N_2_ O_13_	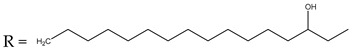	Roots, fruits	[24]

**Table 2 plants-13-00771-t002:** *In vitro* and *in vivo* effects and mechanisms of *Withania somnifera* (L.) Dunal derivatives related to neurodegenerative diseases.

Extract/Compound	Assay/Model	Treatment Time	Doses/Concentration	Biological Activity	Ref.
**ANTI-ALZHEIMER ACTIVITY**					
Methanol root extract of *W. somnifera*	SK-N-SH cells	5 days	5 μg/mL	↑MAP2 ↑PSD-95 ↑neurite outgrowth	[45]
Methanol root extract of *W. somnifera* Withanolide A Withanoside IV Withanoside VI	SH-SY5Y cells	6 days	5 μg/mL 1 µM	↑neurite outgrowth	[46]
Aqueous root extract of *W. somnifera*	A*β* (50 μg/mL) peptide co-incubated with cholesterol (0.2 mg/mL) A*β* (5 μg/mL) PC12 cells + H_2_O_2_ (200 µM)	24 h 24 h	50 μg/mL 6.25–50 μg/mL 50–100 μg/mL	↓A*β* fibrillogenesis ↓H_2_O_2_ cytotoxicity ↓A*β*1-42-induced cytotoxicity	[59,63]
Withanolide A Withanolide B Withanoside IV Withanoside V	SK-N-SH cells + 25 μM of A*β* for 6 h	48 h	10, 20, 40, 60, 80, 100 μM	↓fibrillization ↓cytotoxic effects of amyloid *β* fibrillation ↓ROS ↓number of apoptotic cells	[28]
Withaferina A	SH-APP cells and microglial cells (CHME5)	24 h 48 h	50 nM–1 μM 1 μM	↓A*β*40 ↓NF-kB	[65]
Withanolide A	primary rat cortical neurons (rat pups)	24 h	100 μM	↓BACE1 ↑ADAM10 ↑IDE	[50]
Withanamides A (WA) and Withanamides C (WC) (methanolic fruits extract)	PC-12 cells + *β*-amyloid (25–35)	48 h	50 μg/mL	↓cell death ↓BAP	[32]
Withanoside IV (methanol root extract) Sominone Withanolide A (root of *W. somnifera*)	*In vivo:* Male ddY mice A*β* (25–35) (25 nmol) injected into the right ventricle (i.c.v. injection) *In vitro:* rat cortical neurons damaged by 10 µM A*β* (25–35) for 4 days	administered orally once daily for 13 days 7 days	10 μmol/kg/day 1 µM	↓neuritic losses ↓synaptic loses ↑pre- and postsynapses in the neurons ↓memory deficits ↑Axons length ↑Dendrites length ↑numbers of synapses	[22,47]
Withanosides IV and VI Withanolide A	rat cortical neurons	6 days	1 µM 1 µM	↑MAP2-positive neurites ↑Dendrites length ↑NF-H-positive neurites ↑Axons length	[33]
Methanol:Chloroform (3:1) extract of *W. somnifera* Withanolide A	human neuronal SK-N-MC cell line + *β*-amyloid 1-42	cells were pre-incubated with the extract for 3 h and then exposed to A*β* (5 µM) for 24 h	0.15 µg/mL 1.25 µg/mL–20 µg/mL	↓cytotoxic effects of *β*-amyloid ↓amyloid-*β* 1–42 Internalization ↓MAP2 ↓AChE activity	[54,55]
Withaferin A	SH-APP cells	48 h	0.5–10 μM results showed that 2 μM of Withaferin A reduces the secreted A*β*40	↓A*β*40	[64]
Withanone (WS-2)	Male Wistar rats, 20–24 weeks old with a weight range of 320–360 g and female balb/c mice, 10–12 weeks old weighing 24–28 g	21 days	5, 10 and 20 mg/kg p.o. (most effective dose being 20 mg/kg)	↓amyloid β-42 ↓TNF alpha ↓IL-1 beta, IL-6, MCP-1, Nitric oxide, lipid peroxidation ↓*β*- and *γ*-secretases ↑Ach	[31]
Fresh leaf juice	*In vivo:* Adult albino mice (45 days old, body mass 25 ± 5 g) of both sexes *In vitro*: Primary hippocampal cell cultures of mice pups		daily single doses (4 mL/kg)	↓AchE activity ↓NADPH-d	[82]
Methanol and aqueous extracts from *W. somnifera* (root)	Adult male Wistar rats, 4 months of age (300–350 g) + Scopolamine: (2 mg/Kg b.w. i.p.)	10 days	methanol extract: 100, 200 and 300 mg/Kg b.w. oral aqueous extract: 100, 200 and 300 mg/Kg b.w. oral Donepezil (a reference control) 5 mg/Kg b.w. oral	↓AChE activity	[84]
Sominone	*In vitro*: Rat cortical neurons *In vivo*: Male ddY mice	4 days	0.001–10 µM (max dose 0.1 µM for axon extension and 1 µM for dendrite extension) 10 µmol/kg (injected i.p.)	↑Axons length ↑Dendrites length ↑spatial memory ↑phosphorylation level of RET	[49]
*W. somnifera* roots methanolic extract	Male 5xFAD mice (1-month-old, weighing 16.5 ± 2.6 g) The mice were divided into 4 groups of 12 in each and were treated with intra-gastric gavage daily for 45 days (P30-75)	45 days	200–400 mg/kg/day Barnes circular maze task and Y-maze spontaneous alternation task test	↑expression of NCX3 in the cortex and hippocampus ↓A*β*1–42 aggregation ↑spatial working memory	[52]
Sominone	APP/PS1 doubly transgenic mice (5XFAD mice, Tg6799 line) Rat (SD, embryonic day 18) cortical neurons + 5 μM A*β*(1–42) for 3 days	5 days	10 μmol/kg/day, i.p.	↑axonal density ↑recognition memory	[48]
Partially purified *W. somnifera* extract consisting of 75% withanolides and 20% withanosides	male APP/PS1 mice (middle-aged 9–10 month old)	30 days	1 g/kg body weight o.s.	↑levels of sLRP in plasma ↑expression of LRP and NEP in the liver ↑plasma A*β*42/40 ↓brain A*β* monomer levels	[74]
Alcoholic extract of *W. somnifera* leaves (i-Extract)	Young (12 ± 2 weeks) male Swiss albino strain mice + 3 mg/kg bw scopolamine hydrobromide	7 days	200 mg/kg bw i-Extract dissolved in 0.05% DMSO	↓AChE mRNA level ↓AChE activity ↑ChAT activity ↑ChAT mRNA ↑Arc protein	[80]
I-Extract and whitanone	Male Swiss albino strain mice of 12 weeks (25–30 g) + intraperitoneally injected with scopolamine 3 mg/kg BW Glioma C6 (rat) and neuronal IMR32 (human) cell lines + scopolamine (3 mM) was added to the cells for 1 to 2 h	7 days 24 h	i-Extract (100 mg, 200 mg, 300 mg/kg BW) i-Extract, 0.8 mg/mL and withanone 5 mg/mL	↑neurite and axon growth ↑MAP-2 ↑GAP-43 protein ↑NF-H ↑PSD-95 ↓apoptosis ↑BDNF ↑GFAP	[79]
Methanolic and water extracts of *W. somnifera* root	Wistar strain male albino rats having body weights 300–350 g + scopolamine (2 mg/Kg body wt) injected intraperitoneally	30 days	methanolic extract 300 mg/Kg body wt; water extract (300 mg/Kg body wt); donepezil hydrochloride (a standard control) 5 mg/Kg body wt	↓glutamate levels ↑GDH activity	[142]
Withanolide-A	Neuro2a mouse neuroblastoma cell line	24 h	pre-treatment with 2.5, 5, 10, and 20 μM of withanolide-A and then exposition to 10 mM glutamate for 2 h	↓apoptosis ↓Intracellular calcium ↓ROS ↓p53 ↓Bax/Bcl-2 ratio	[88]
Water extract of leaves of Ashwagandha (ASH-WEX)	C6 and IMR-32 cells	24 h	Pretreatment with ASH-WEX (0.1%) 0.5 mM–1 mM glutamate	↓apoptosis ↓GFAP ↓GFAP mRNA ↑HSP70 ↓MMP2 e MMP9 ↑NCAM	[89]
Withanone	differentiated Neuro2a cells	1 h	pre-treated with 5, 10 and 20 μM doses of Withanone and then exposed to 3 mm NMDA	↓apoptosis ↓Bax/Bcl-2 ratio ↓Intracellular calcium ↓ROS ↓cytochrome c ↑pro caspase-3 ↑pro caspase-9 ↓cleaved caspase-3 ↓MDA	[86]
I-Extract (alcoholic extract of *W. somnifera* leaves)	*In vivo:* Male Swiss albino strain mice (8 ± 1 weeks) *In vitro*: pimary hippocampal neurons	7 days	200 mg/kg o.s. Scopolamine Hydrobromide 3 mg/kg	↑KLK8 ↑MAP2	[81]
*W. somnifera* alcoholic extract (withaferin-A)	Male Wistar rats (1 year old) weighing 400 ± 20 g	3 weeks	Pre-treatment: 100–200–300 mg/kg bilateral ICV injection of STZ (3 mg/kg body weight in saline, 5 μL/injection)	↓cognitive deficits ↓AChE activity ↑ChAT expression ↓caspase-3 activity	[143]
**ANTI-PARKINSON ACTIVITY**					
*W. somnifera* (roots) methanolic extracts	*Caenorhabditis elegans* BZ555 and NL5901 strains		66 mg/mL	↓*α*-synuclein aggregation	[113]
Standardized root methanolic extract of *W. somnifera* (Wse)	Adult wild type (WT) LRRK2 WD40 mutant *Drosophila melanogaster* (Dm) males.		0.1, 1 and 10% *w*/*w*	↑lifespan of mutants ↓time to climb ↓mutation-related loss of mitochondrial structural integrity	[97]
Water root extract *of W. somnifera* (KSM-66 extract)	human neuroblastoma SH-SY5Y cell line + 6-OHDA (3.9–2000 μM)	24 h	0.25–10 mg/mL (KSM-66 extract was cytotoxic at concentrations greater than 1 mg/mL)	↑cell survival ↑thioltransferase activities ↑peroxidase activity ↑peroxiredoxin-1 ↑VGF ↓vimentin ↓protein glutathionylation ↑intracellular ATP levels ↓mitochondrial dysfunction	[98]
Ethanolic crude extract of *W. somnifera*	Male Wistar rats (weighing 150–200 g) + 6-OHDA	3 weeks	100–200–300 mg/kg o.s.	↓TBARS ↑GSH ↑GST ↑GPx ↑SOD activity ↑CAT activity ↓D2 receptor ↑DA, DOPAC and HVA	[99]
Ethanolic root extract	Male Swiss albino mice weighing 25 ± 5 g + MB (30 mg/kg b.w., intraperitoneally) and PQ (10 mg/kg b.w., intraperitoneally)	3–6–9 weeks	100 mg/kg body weight, orally daily	↓walking errors ↑Stride length ↓MDA level ↓levels of nitrite ↑level of catalase ↑TH-positive dopaminergic neurons ↑motor coordination ↑dopamine, DOPAC, and HVA levels ↑Bcl-2 protein ↓Bax ↓iNOS and iNOS mRNA ↓GFAP expression	[100,101]
Ethanolic root extract 1:10 (*w*/*v*)	male Long-Evans hooded rats + received 5 intraperitoneal injections of PQ dissolved in 1 × phosphate-buffered saline (PBS) at 10 mg/kg (One injection occurred every 5 days over 20 days)	4 months	2 mg/mL	↑Pro-BDNF ↑TH ↓4-HNE ↑CARP1 ↑GDNF	[119]
Standard root extract of *W. somnifera* powder	Prepubertal male mice (4 wk old, 25 ± 2 g) + ROT (0.5 and 1.0 mg/kg b.w/d for 7 consecutive days, i.p.)	4 weeks	100–200–400 mg/ kg b.w/o.s	in Cerebellum (cb) and Striatum (st): ↓ROS, MDA, ↓hydroperoxides ↓Nitric oxide levels ↑reduced glutathione ↑total thiol content ↑SOD, CAT, GPx ↑GSH ↓AChE activity ↑Stride length	[95]
*W. somnifera* root extract standard powder, (withanolides, 2.57%; withaferin A, 2.38%)	*Drosophila melanogaster* flies (8–10 d) + ROT (IC_50_: 500 μM, 7 d)	5 days	0.005, 0.01, 0.05%	↓Incidence of mortality ↓motor dysfunctions ↓ROS ↓LPO ↓HP (hydroperoxides) ↑NPT (non-protein thiols) ↑GSH ↑SOD ↓AChE activity ↑SDH activity ↑MTT activity ↑dopamine levels ↑complex I–III and complex II–III enzymes	[96]
*W. somnifera* root extract Ashwagandha leaf extract (A-Extract)	Male Albino mice weighed between 30 and 35 g + MPTP 20 mg/kg body weight/day for 4 days i.p.	7–28 days	100 mg/kg body weight	Midbrain and corpus striatum: ↑DA ↑DOPAC ↑HVA ↑GSH ↑GPx ↓TBARS ↓MDA activity ↑motor function ↓SOD activity ↓CAT activity	[103,104,105,106]
Withanolide A (root extract)	C. elegans strain (NL5901)	72 h	5 μM	↓*α*-synuclein levels	[114]
**ANTI-HUNTINGTON ACTIVITY**					
*W. somnifera* root extract	Male Wistar rats (weighing between 250 and 300 g) + 3-NP (10 mg/kg, i.p.) for 14 days	14 days	100–200 mg/Kg o.s	Striatum and cortex: ↑motor activity ↑muscle grip strength ↓MDA ↓nitrite ↓SOD ↑catalase enzyme activity ↓LDH activity ↑mitochondrial enzyme complexes (I, II, and III) ↑memory performance (Morris water maze test and elevated plus maze paradigms) In striatum cortex and hippocampus region: ↑glutathione ↓AChE activity	[124,125]
Withaferin A	HD150Q cells that inducibly expresses truncated N-terminal mutant huntingtin containing 150Q fused with GFP (tNhtt-150Q-GFP) Mouse model of Huntington’s disease (strain B6CBA-Tg (HDexon1) 62Gpb/3 J) (also referred as R6/2 line)	48 h 12 weeks	0.1–0.5 µM 1 mg/kg body weight (15 doses were administered via intra-peritoneal route every alternate day starting at their age of 56th day until the 84th day)	↓tNhtt-150Q-GFP aggregation ↓soluble level of tNhtt-150Q-GFP protein ↑HSP70 ↑body weight ↑average lifespan ↓motor deficits partial restoration of striatal atrophy ↓number of nuclear aggregates in the striatal, cortical and hippocampal areas ↓insoluble mutant huntingtin ↓level of soluble transgenic mutant huntingtin ↓number of activated microglia and Iba1	[126]
*W. somnifera* ethanolic extract withanolide A	adult male Sprague Dawly rats weighing (200–240 g) + Kainic acid was dissolved in 0.9% NaCl. 0.5 micrograms in 0.5 μL of kainic acid		25 mg/kg body weight 100 mg/kg body weight	↓abnormal movement or posture	[127]
**ANTI-ALS ACTIVITY**					
*W. somnifera* root (11:1) extract	Transgenic mice carrying G93A sod1 mutant	50 days	5 mg of root powder (by gavage) as a suspension in 200 μL sterile buffered saline every alternate day beginning at 50 days of age and continued till the mice were capable to ingest.	↑survival ↑motor performance ↓misfolded SOD1 species ↑Hsp-70, Hsp-27 levels ↑motor neurons in the ventral horn of spinal cords ↑motor axons in the ventral roots ↓TLR2 and CD68 levels ↓COX-2 ↓phosphorylation of p65 NF-κB ↓p62 levels	[132]
Withaferin A	SOD1 (G93A) or SOD1 (G37R) transgenic mice		4 mg/kg of body weight i.p. (twice a week)	↑survival ↓the loss of motor function ↓the loss of body weight ↓levels of misfolded SOD1 ↑Hsp25 and Hsp70 ↓Iba-1 and Toll-like receptor 2 expression ↓astrogliosis and microgliosis.	[131]
*W. somnifera* root (11:1 extract from the plant root)	hTDP-43A315T mice (transgenic mice expressing a genomic fragment encoding human TDP-43A315T mutant)	8–16 weeks	5 mg root powder (by gavage) as a suspension in 200 μL sterile buffered saline every alternate day	↑motor and cognitive performance ↑innervated neuromuscular junctions (NMJs) ↓astrogliosis and microgliosis ↓phophorylated p65 NF-κB ↓peripherin levels	[136]
Withaferin A	cultures of primary cortical neurons and microglia from transgenic mice overexpressing TDP-43WT or TDP-43 mutants TDP-43WT; GFAP-luc double transgenic mice	24 h 10 weeks	1 μM 3 mg/kg body weight twice a week i.p.	↓cell death ↓GFAP-luc expression in the spinal ↓p65 levels ↑motor performance ↓peripherin aggregates ↓activated microglia ↓number of partially denervated neuromuscular junctions (NMJs)	[130,135]

↓ reduction, ↑ increase.

**Table 3 plants-13-00771-t003:** Checklist for assessment of risks of bias in pre-clinical studies [144].

Checklist for Assessment of Risks of Bias in Pre-Clinical Studies
Are the hypothesis and objective of the study clearly described?
Are the main outcomes to be measured clearly described?
Are the main findings of the study clearly described?
Are the samples size calculations reported?
Are the animals randomly housed during the experiment?
Are the investigators blinded from knowledge which treatment used?
Are the outcome assessors blinded?
Is the dose/route of administration of the *Withania somnifera* (L.) Dunal properly reported?
Is the dose/route of administration of the drug in co-treatment properly reported?
Is the frequency of treatments adequately described?

**Table 4 plants-13-00771-t004:** Checklist for assessment of risks of bias in clinical studies [144].

Checklist for Assessment of Risks of Bias in Clinical Studies
Was the sample frame appropriate to address the target population?
Were study participants sampled in an appropriate way?
Was the sample size adequate?
Were the study subjects and the setting described in detail?
Was the data analysis conducted with sufficient coverage of the identified sample?
Were valid methods used for the identification of the condition?
Was the condition measured in a standard, reliable way for all participants?
Was there an appropriate statistical analysis?
Was the response rate adequate, and if not, was the low response rate managed appropriately?

## Data Availability

Not applicable.

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
