# Peer review of "Withania somnifera (L.) Dunal, a Potential Source of Phytochemicals for Treating Neurodegenerative Diseases: A Systematic Review"

_plants, 2024, doi:10.3390/plants13060771_

Round 1

Reviewer 1 Report

Comments and Suggestions for Authors

The topic discussed in this work is very important in view of the overuse of drugs and their reduced effectiveness. The authors conducted a proper analysis of the usefulness of the available literature, and the selected literature data allowed for a very broad but at the same time in-depth analysis of the topic. The work is very interesting and is a valuable source of data. The authors rightly point out the lack of clinical research on Ashwaghanda, which has already been carried out, among others, in dietary supplements.

Author Response

We thank the reviewer; we are glad you liked our work.

Reviewer 2 Report

Comments and Suggestions for Authors

This is a well written and comprehensive review of preclinical and clinical studies relating to the potential role of Withania somnifera (ashwagandha) and some of its withanolide components in ameliorating neurodegenerative diseases.

The extensive data has been presented in a logical manner, and in both long form text and summarized in Table form.   While this repetition could be considered unnecessary, it is actually helpful to the reader. The table provides an easily accessible summary where comparison of studies is facilitated, while the text explains the results in the context of the pathological features of each neurological disease.  At the same time,  too much fine experimental detail is given in the text (dose, duration of treatment).  Please remove unnecessary details from the text since they are also given in the table.  This will also make the flow of reading better.

Some other specific recommended changes are:

Title and elsewhere:  The correct Latin binomial with botanist authority is Withania somnifera (L.) Dunal  not “Withania somnifera L.”

Every use of Withania somnifera or W. somnifera should be italicized. There are numerous incidences where it is not.

Other Latin names like Drosophila melanogaster and Caenorhabditis elegans should also be italicized.

Abstract Line 18:  add the word “potential”  - “of great interest is W. somnifera’s potential beneficial effect..

Abstract lines 22-23  - should add that ashwagandha extracts and compounds diminish abnormal protein aggregations seen in preclinical models of neurodegenerative disease.

Line 49-50:  clarify that W somnifera is “potentially” useful as use for these disease conditions has not been clinically proven.

Line 77-82:  References need to be given for these claims or at least say “as shown in rest of this article…”

Figure 1:   Too small to read comfortably.  Please increase size and clarity

Table 1:  Indicate in a footnote that the neuroprotective activities can be found in Table 2.

Table 1:  Withanamide is spelled wrong as Whitanamide

Line 114-115:  The text in the cited reference is “The characteristic feature of withanolides and ergosane-type steroids is one C8 or C9-side chain with a lactone or lactol ring but the lactone ring may be either six-membered or five-membered and may be fused with the carbocyclic part of the molecule through a carbon-carbon bond or through an oxygen bridge.   Please confirm that the possibility of an oxygen bridge applies to withanolides.  The original paper may have been referring to other ergosane-type steroids.  None of the withanolides shown have an oxygen bridge.

Line 134:  correct spelling to “Withanamides”

Line 141:   The reference [32] given to support the statement “Withanamide A has been shown to be effective in treating Alzheimer’s Disease”  is an in vitro study in PC12 cells!!!  Please change the statement to match the content of the reference.  “shown to be effective in treating Alzheimer’s Disease” implies a clinical trial!

Line 192 -197:  in describing the method and “inclusion criteria”  please refer to the methods section number where these details can be found.

Figure 4: change “argument” to “subject area”

Line 208:   correct spelling and Latin name

Line 209:  change “first” to “leading”

Line 237and onwards:  Please confirm it is Withanoside VI (six).  Withanoside V is more common.

Line 255 to 256:   The withanosides are glycosides.  Its not correct to say they have glycosides in their structure. 

Line 258-262;   Too much fine experimental detail given. Interested readers can look up the original paper. Just give the broad picture.

Figure 5:  Please increase size and clarity

Line 318 to 321:  what species/model was this study done in?

Line 361-363:  the biggest difference between withanolides and withanosides is that withanoside have a 2-sugar chain which adds 12 carbons. The sentence calling them C-40 steroids is misleading.

Line 390:  should dPC12 be PC12?

Line 410:  What is LVFFA? Not defined.

Line 422 onwards:  a new term BAP is used for beta amyloid protein. Better stick to the terminology used earlier in the review. BAP (25-35) seems to be the same as Abeta (25-35).

Line 500:  Section 2.2.4 – why is this section placed separately from earlier sections on interaction of ashwagandha with Abeta?  Can it be combined there?

Line 663-667:  Description of two different studies have been combined here and its confusing. Please rewrite.

Line 756, section 2.3.5:   Add what W. somnifera is synergizing with

Line 839:   misspelling witanolide A

Line 911 – 925:   Please use a consistent way to denote the hTDP43 transgenic mice. Various terms are used within the same paragraph.

Table 2:  Change title to “In vivo and in vivo effects and mechanisms of Withania somnifera derivatives related to neurodegenerative diseases.

Table 2:   Consistency needed in use upper or lower case for extract/compound

Table 2:   Anti SLA activity – change to Anti ALS activity

Conclusion:  Include something about the potential to influence abnormal protein deposition in the various neurodegenerative diseases.

Conclusion:  change last word from “demanded” to “required”.

Author Response

The extensive data has been presented in a logical manner, and in both long form text and summarized in Table form.   While this repetition could be considered unnecessary, it is actually helpful to the reader. The table provides an easily accessible summary where comparison of studies is facilitated, while the text explains the results in the context of the pathological features of each neurological disease.  At the same time,  too much fine experimental detail is given in the text (dose, duration of treatment).  Please remove unnecessary details from the text since they are also given in the table.  This will also make the flow of reading better.

Auth: We thank for the suggestion; most of the unnecessary details has been removed

Some other specific recommended changes are:

Title and elsewhere:  The correct Latin binomial with botanist authority is Withania somnifera (L.) Dunal  not “Withania somnifera L.”

Auth: We thank for the correction, we have taken care to correct it

Every use of Withania somnifera or W. somnifera should be italicized. There are numerous incidences where it is not.

Auth: We thank for the correction, we made sure to put everything in italics

Other Latin names like Drosophila melanogaster and Caenorhabditis elegans should also be italicized.

Auth: We thank for the correction, we provided to made them in italics

Abstract Line 18:  add the word “potential”  - “of great interest is W. somnifera’s potential beneficial effect..

Auth: We thank for the suggestion, we have provided to make the word potential

Abstract lines 22-23  - should add that ashwagandha extracts and compounds diminish abnormal protein aggregations seen in preclinical models of neurodegenerative disease.

Auth: We thank for the suggestion, we have provided to add this phrase in the abstract

Line 49-50:  clarify that W somnifera is “potentially” useful as use for these disease conditions has not been clinically proven.

Auth: We thank for the suggestion, we have provided to make the word potential

Line 77-82:  References need to be given for these claims or at least say “as shown in rest of this article…”

Auth: We thank for the suggestion, we have provided to insert this phrase

Figure 1:   Too small to read comfortably.  Please increase size and clarity

Auth: We thank for the suggestion, we have tried to increase the size of the figure

Table 1:  Indicate in a footnote that the neuroprotective activities can be found in Table 2.

Auth: We thank for the suggestion, we have inserted the footnote

Table 1:  Withanamide is spelled wrong as Whitanamide

Auth: We thank the reviewer we have corrected  it

Line 114-115:  The text in the cited reference is “The characteristic feature of withanolides and ergosane-type steroids is one C8 or C9-side chain with a lactone or lactol ring but the lactone ring may be either six-membered or five-membered and may be fused with the carbocyclic part of the molecule through a carbon-carbon bond or through an oxygen bridge.”   Please confirm that the possibility of an oxygen bridge applies to withanolides.  The original paper may have been referring to other ergosane-type steroids.  None of the withanolides shown have an oxygen bridge.

Auth: We thank the reviewer we have corrected  it

Line 134:  correct spelling to “Withanamides”

Auth: We thank the reviewer we have corrected  it

Line 141:   The reference [32] given to support the statement “Withanamide A has been shown to be effective in treating Alzheimer’s Disease”  is an in vitro study in PC12 cells!!!  Please change the statement to match the content of the reference.  “shown to be effective in treating Alzheimer’s Disease” implies a clinical trial!

Auth: We thank the reviewer we have corrected  it

Line 192 -197:  in describing the method and “inclusion criteria”  please refer to the methods section number where these details can be found.

Auth: We thank for the suggestion, we have inserted it

Figure 4: change “argument” to “subject area”

Auth: We thank the reviewer we have corrected  it

Line 208:   correct spelling and Latin name

Auth: We thank the reviewer we have corrected  it

Line 209:  change “first” to “leading”

Auth: We thank the reviewer we have corrected  it

Line 237and onwards:  Please confirm it is Withanoside VI (six).  Withanoside V is more common.

Auth: We confirm it

Line 255 to 256:   The withanosides are glycosides.  Its not correct to say they have glycosides in their structure. 

Auth: We thank the reviewer we have corrected  it

Line 258-262;   Too much fine experimental detail given. Interested readers can look up the original paper. Just give the broad picture.

Auth: We thank the reviewer we have corrected  it

Figure 5:  Please increase size and clarity

Auth: We thank for the suggestion, we have tried to increase the size of the figure

Line 318 to 321:  what species/model was this study done in?

Auth: We thank for the suggestion, we have specified it

Line 361-363:  the biggest difference between withanolides and withanosides is that withanoside have a 2-sugar chain which adds 12 carbons. The sentence calling them C-40 steroids is misleading.

Auth: This is correct

Line 390:  should dPC12 be PC12?

Auth: Yes we have correct it

Line 410:  What is LVFFA? Not defined.

Auth: We thank for the suggestion, we have tried to increase the size of the figure

Line 422 onwards:  a new term BAP is used for beta amyloid protein. Better stick to the terminology used earlier in the review. BAP (25-35) seems to be the same as Abeta (25-35).

Auth: We thank the reviewer we have corrected  it

Line 500:  Section 2.2.4 – why is this section placed separately from earlier sections on interaction of ashwagandha with Abeta?  Can it be combined there?

Auth: We thank for the suggestion, but we prefer to maintain them separatelly

Line 663-667:  Description of two different studies have been combined here and its confusing. Please rewrite.

Auth: We thank for the suggestion, we have included references to make it clearer

Line 756, section 2.3.5:   Add what W. somnifera is synergizing with

Auth: We thank for the suggestion, we have specified it

Line 839:   misspelling witanolide A

Auth: We thank the reviewer we have corrected  it

Line 911 – 925:   Please use a consistent way to denote the hTDP43 transgenic mice. Various terms are used within the same paragraph.

Auth: We thank for the suggestion, we have correct it

Table 2:  Change title to “In vivo and in vivo effects and mechanisms of Withania somnifera derivatives related to neurodegenerative diseases.

Auth: We thank for the suggestion, we have changed the table title

Table 2:   Consistency needed in use upper or lower case for extract/compound

Auth: We thank for the suggestion, we have correct it

Table 2:   Anti SLA activity – change to Anti ALS activity

Auth: We thank for the suggestion, we have correct it

Conclusion:  Include something about the potential to influence abnormal protein deposition in the various neurodegenerative diseases.

Auth: We thank for the suggestion, we have inserted it

Conclusion:  change last word from “demanded” to “required”.

Auth: We thank for the suggestion, we have changed it